# Freezing of few nanometers water droplets

Alireza Hakimian[1], Mohammadjavad Mohebinia [2], Masoumeh Nazari[1], Ali Davoodabadi[1], Sina Nazifi[1], Zixu Huang[1], Jiming Bao[2] & Hadi Ghasemi [1,3 ✉]

Water-ice transformation of few nm nanodroplets plays a critical role in nature including climate change, microphysics of clouds, survival mechanism of animals in cold environments, and a broad spectrum of technologies. In most of these scenarios, water-ice transformation occurs in a heterogenous mode where nanodroplets are in contact with another medium. Despite computational efforts, experimental probing of this transformation at few nm scales remains unresolved. Here, we report direct probing of water-ice transformation down to 2 nm scale and the length-scale dependence of transformation temperature through two independent metrologies. The transformation temperature shows a sharp length dependence in nanodroplets smaller than 10 nm and for 2 nm droplet, this temperature falls below the homogenous bulk nucleation limit. Contrary to nucleation on curved rigid solid surfaces, ice formation on soft interfaces (omnipresent in nature) can deform the interface leading to suppression of ice nucleation. For soft interfaces, ice nucleation temperature depends on surface modulus. Considering the interfacial deformation, the findings are in good agreement with predictions of classical nucleation theory. This understanding contributes to a greater knowledge of natural phenomena and rational design of anti-icing systems for aviation, wind energy and infrastructures and even cryopreservation systems.

[1] Department of Mechanical Engineering, University of Houston, 4726 Calhoun Rd, Houston, TX 77204, USA. [2] Department of Electrical and Computer Engineering, University of Houston, 4726 Calhoun Rd, Houston, TX 77204, USA. [3] Department of Chemical and Biomolecular Engineering, University of Houston, 4726 Calhoun Rd, Houston, TX 77204, USA. ✉email: hghasemi@uh.edu

Climate fluctuations, global radiative properties, and microphysical processes in clouds are strong functions of the water nucleation mechanism and distribution of ice particles. Ice particles formed by various mechanisms in different shapes have critical roles in hydrological fluxes and the climate[1,2]. Ice nucleation in the atmosphere could occur either on particles (heterogeneous) or in a liquid droplet (or a dilute solution droplet) surrounded by a vapor environment (homogeneous). The required number of water molecules to form crystalline ice from the liquid phase is determined to be 275 through infrared (IR) excitation-modulated photoionization spectroscopy (~2 nm droplet)[3]. Homogeneous freezing of nanoscopic liquid water droplets is explored through hypersonic expansion and the freezing temperature is demonstrated[4,5] to occur in "no man's land"[6]. Also, computational approaches[7–9] thorough large-scale molecular dynamics (MD) simulation have highlighted the suppression of ice nucleation in nanodroplets.

On the other hand, heterogeneous ice nucleation occurs in the atmosphere, a wide range of species (e.g., the extracellular matrix in wood frogs[10], Antarctic fishes[11,12]) and technological embodiments (e.g., aviation[13] and infrastructures, including transportation, power, and energy systems[14–16]) and causes outstanding financial burden[17]. Nanoscopic anti-icing surfaces developed to address the icing challenge aim to tune the water–ice transformation at a few nanometer scales. These include nanostructured surfaces[18–20], slippery liquid-infused porous surfaces[21,22], magnetic slippery surfaces[23], and even anti-frosting[24–28] surfaces. Furthermore, confinement effects could drastically affect the phase-change phenomenon in membrane[29], porous material[30–33], nanochannels[34,35], and carbon nanotubes[36,37]. However, direct probing of water–ice phase transformation in a few nanometer scales in heterogeneous environments has been challenging: nanoscopic water droplets could evaporate or grow by condensation extremely fast (i.e., order of $10^{-35}$ s)[38]. Hence, the study of nanodroplets in confinements is needed to accurately probe their phase transformation.

Here, we report water–ice phase transformation in confined geometries down to 2 nm in diameter (Fig. 1a). It is shown that despite being heterogeneous, ice nucleation in a 2 nm length-scale occurs at a lower temperature than homogeneous bulk nucleation (see Fig. 1a). We form water nanodroplets with diameters ranging from 150 to 2 nm in membrane pores surrounded by an oil environment (Fig. 1b). The high interfacial curvature of these nanodroplets leads to large positive pressures in these droplets (up to 500 bar). Through two independent electrical resistance metrology and Fourier-transform infrared spectroscopy (FTIR) spectroscopy, the ice nucleation temperature in these nanodroplets is probed. The phase transformation temperature of nanodroplets has a length dependence and this dependence becomes more pronounced at the sub-10 nm scale. The ice phase formed within nanodroplets as small as 2 nm in diameter is possibly stacking default ice (Isd) and could transition to hexagonal ice (Ih) in a slow kinetically controlled manner. At a few nanometer scales, the softly curved interface of oil–water plays a critical role in the suppression of ice nucleation and the characteristics of this interface are entirely different than those of concave stiff solid-water interfaces.

## Results

The water nanodroplets are formed inside pores of anodized aluminum oxide (AAO) membranes. A range of AAO membranes with pore diameters of 150–2 nm, membrane diameter of 1 cm, and membrane thickness (i.e., pore length) of 50–60 μm are acquired. The surface geometry of these pores is probed by scanning probe microscopy as shown in Supplementary Fig. 1. For all membranes, the pore diameters are highly uniform, but for 2 nm membrane, pore dimensions have a distribution of 2–4 nm. In the exception of 5 and 2 nm membranes, the pores are extended uniformly along the thickness of the membrane and are called isotropic. The 5 and 2 nm membranes are anisotropic and are made of two layers with different pore dimensions: small pores are extended from one side to approximately 10% of the thickness of the membrane (active layer) and the remainder of the membrane is made of pores with a dimension of 150 nm (supporting layer). The procedure for the formation of nanodroplets in these nanopores is discussed in Methods (Confined nanodroplets). These nanodroplets are surrounded by an oil environment forming an oil–water interface. The criteria for selection of the oil were (1) to wet the pore wall and (2) to maximize water–oil interfacial tension. Octane was selected as the most appropriate oil to form nanodroplets in the pores after various choices of the oil were compared as discussed in Supplementary Note 2. To ensure the presence of water nanodroplets in the pores, electrical resistances across the porous membrane were compared for cases of pores filled with octane and pores with confined water nanodroplets surrounded by octane as shown in Supplementary Note 3. For the case that the pores are completely filled by octane, the electrical resistance is almost 5.5 GΩ. By the introduction of water droplets inside the pores, the electrical resistance is reduced to almost 3 GΩ. Also, the existence of water inside the pores is confirmed with FTIR that will be discussed later. Once the existence of nanodroplets in the pores was confirmed, we studied the water–ice phase change of these nanodroplets through two independent metrologies. Note that ice nucleation in these pores takes place heterogeneously (see Supplementary Note 7), i.e., nucleation initiates from the oil–water interface due to the lower energy barrier required compared to

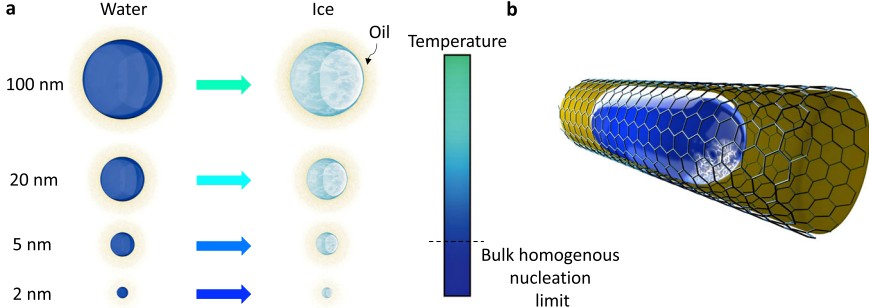

**Fig. 1 Freezing of water nanodroplets. a** The length dependence of water–ice transformation in a heterogeneous mode. For 2 nm water droplets, heterogeneous nucleation could break the limit of bulk homogeneous nucleation. **b** Schematic of a nanodroplet formed in confined geometry and surrounded by an oil environment where ice nucleation occurs at the soft oil–water interface. The ice nucleation changes the local curvature of the oil–water interface.

that for homogeneous nucleation. Also, solid–liquid phase-change temperature of octane is −57 °C, well below the temperatures considered in this study. Once, the entire system including water droplet and surrounding environment is cooled down in a quasi-equilibrium condition, the ice formation is characterized by median ice nucleation temperature ($T_N$)[16]. As the electrical conductance of water and ice are different up to three orders of magnitude[39] (depending on ion concentration), water–ice phase change in nanodroplets should manifest itself in the electrical conductance metric (Supplementary Note 4). Thus, we used electrical conductance metrology across the pores with the four-probe method[40] (that minimizes the effect of parasitic resistances) to investigate phase transformation temperature in water nanodroplets. Fifty parts per million (p.p.m.) of NaCl was added to deionized water to enhance the electrical conductance contrast between water and ice phases. This concentration of salt does not have any measurable effect on phase-change temperature[41]. Also, 250 p.p.m. of Span80 (nonionic surfactant, Sigma-Aldrich) was added to the octane to enhance its electrical conductance. This surfactant did not show any significant effect on oil–water interfacial tension as is shown in Supplementary Note 2. The experimental setup is shown in Supplementary Note 4 and the methodology is presented in Methods II. For a given pore dimension, in a quasi-steady process, we gradually decreased the temperature of the system (0.3 °C/min) and measured the current–voltage ($I$–$V$) curves at each selected temperature, as shown in Fig. 2. As shown, for a representative membrane of 150 nm, as the system temperature decreases, the resistance across the pore increases linearly indicating the effect of temperature on the electrical resistivity of the liquid in the pore. However, between −9 and −11 °C, we detected a high, nonlinear shift in resistivity and a jump in electrical resistance. As

the conductance of ice and water are significantly different, this jump is an indication of water–ice phase transformation inside the pores. We continued these measurements to lower temperatures and no additional jump was detected at lower temperatures. This metrology was conducted on confined nanodroplets in membranes with pores of 80, 40, 20, and 10 nm, and for all these pores, the jump in electrical resistance was observed but at different temperatures indicating the size dependence of $T_N$ of nanodroplets. The results are presented in Supplementary Note 5. As discussed, for membranes with pore dimensions of 5 and 2 nm, in addition to small pores, there are larger pores in the supporting layer with the dimension of 150–200 nm. As we conducted the phase transformation studies on these pores, we observed two jumps, one at higher temperatures (~−8 °C) was associated with larger pore dimensionz and the one at lower temperatures was attributed to phase change within smaller pores. The electrical resistivity results for different pore dimensions along with observed jumps are depicted in Fig. 2e. The size dependence of $T_N$ for nanodroplets of 10 nm and below becomes more pronounced. Interestingly, despite being heterogeneous nucleation, see Supplementary Note 7, we observed that water–ice phase transformation at pore dimension of 2 nm occurs at lower temperatures (−41 °C) than that for *homogeneous bulk nucleation*, (~−38 °C)[42,43]. That is, for a few nanometer water droplets ice formation could be suppressed to extremely low temperatures. We continued these studies with another independent approach to remeasure $T_N$ and acquire an understanding of the type of ice phase formed by these nanodroplets.

In the second metrology, once the nanodroplets are formed in the pores, we probed FTIR spectrum of these droplets as a function of temperature. The detail of this metrology is presented in Methods III. Figure 3a shows FTIR spectroscopy of

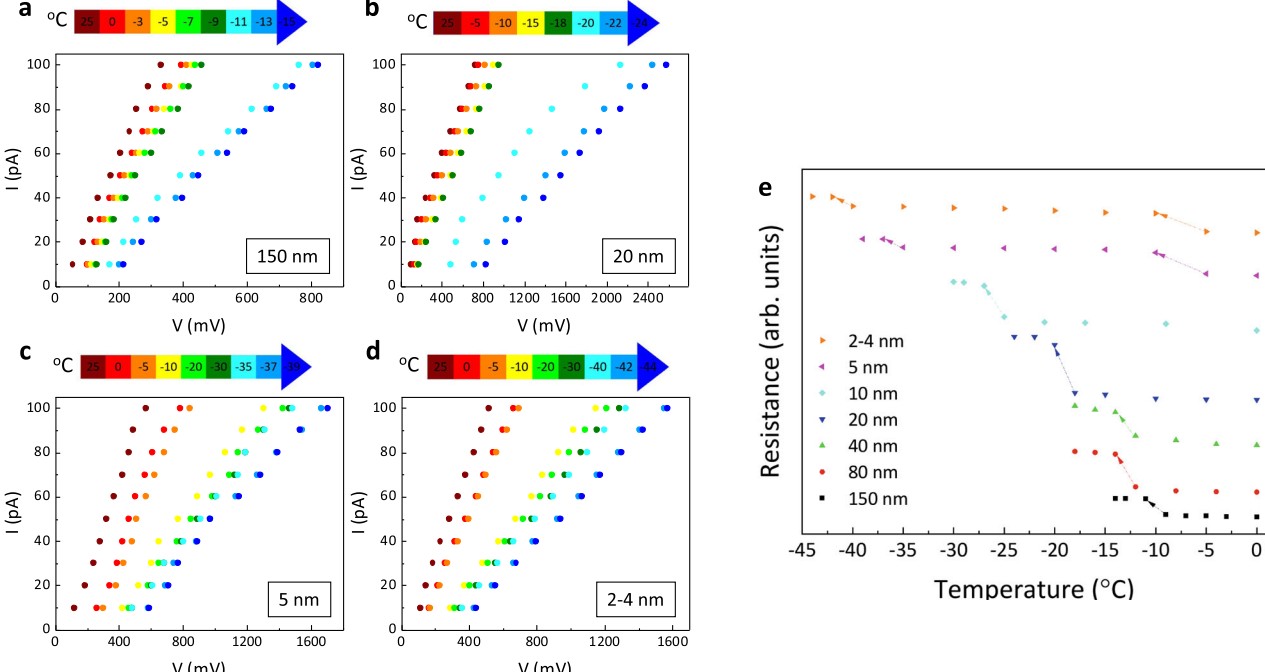

**Fig. 2 Electrical resistance metrology.** *I–V* curves were measured at different temperatures across the nanopores confining nanodroplets. **a** Nanodroplets of ~150 nm, in which a nonlinear jump in electrical resistance is measured between the temperature of −9 and −11 °C. The resistance jump indicates water–ice phase change and suggests nucleation temperature ($T_N$) at this length-scale. **b** For nanodroplets of ~20 nm, the resistance jump and consequently phase change is detected between −18 and −20 °C. **c** The $T_N$ value drops to ~−36 °C for 5 nm nanodroplets. Note that there are two resistance jumps for these pores as there are two dimensions of pore in these membranes, large pores of (150–200 nm) and small pores of 5 nm. **d** The $T_N$ value drops to −41 °C below the limit of homogeneous bulk nucleation for 2 nm water droplets. **e** The normalized resistance jumps for different dimensions of water nanodroplets are shown. The electrical resistance metrology for other pore dimensions is provided in the Supplementary information.

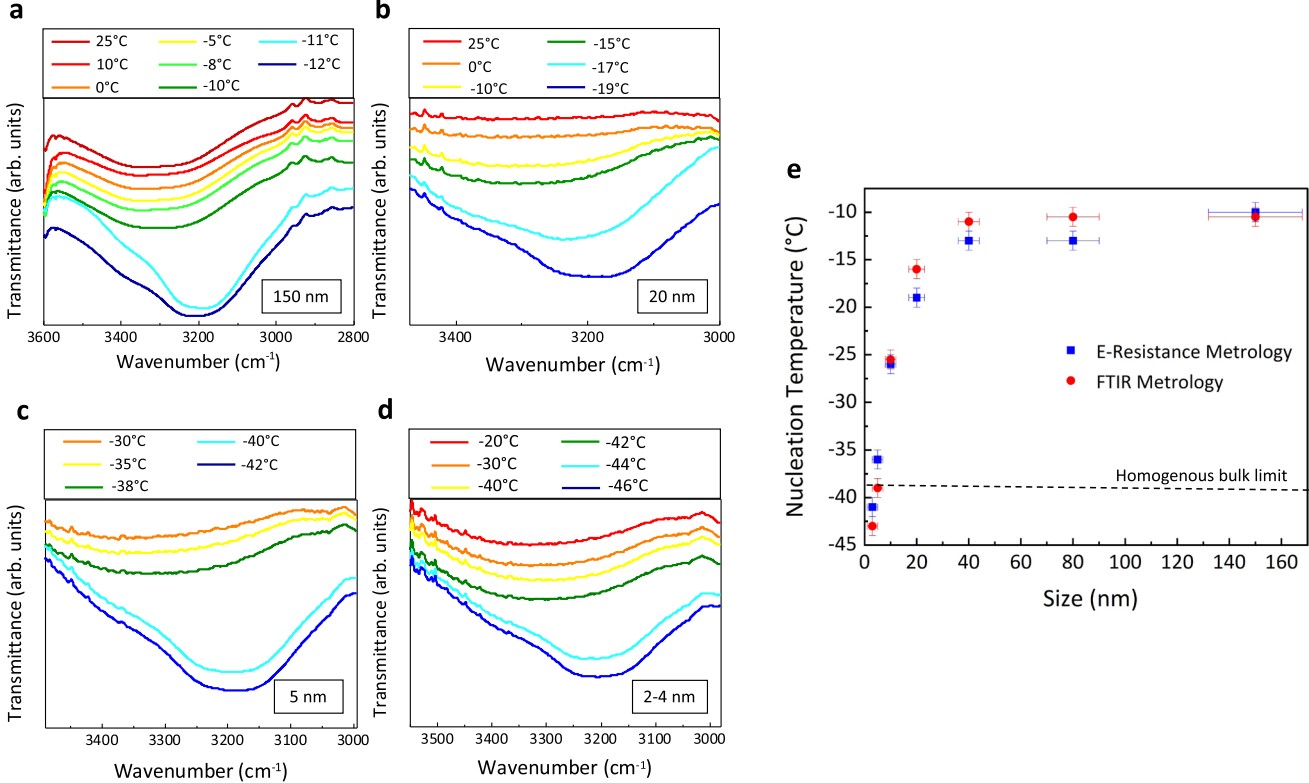

**Fig. 3 FTIR metrology.** FTIR spectrum of nanodroplets in various pore dimensions. **a** For 150 nm nanodroplets, at the temperature of −11 °C, the OH stretch peak is narrowed and red-shifted to a wavenumber of ~3200 cm⁻¹. This redshift indicates the water–ice phase change and provides $T_N$ value at this length-scale. **b** The redshift occurs at a temperature of −17 °C for 20 nm water droplets. **c** For 5 nm water droplets, the narrowing of OH stretch peak occurs at a temperature of −40 °C, while the redshift goes to lower wavenumbers compared to those of larger nanodroplets (150, 80, 40, 20, and 10 nm). This suggests a shorter O–H bond due to the confinement effect for the ice formed. **d** The $T_N$ value for 2 nm droplets becomes ~−44 °C. A similar confinement effect on the shortening of O–H bonds is observed for these droplets. That is for nanodroplets smaller than 5 nm, the formed ice phase has a different bond length compared to the bulk ice. **e** The nucleation temperature of nanodroplets measured through two independent metrologies is shown as a function of length-scale. For nanodroplets of 2 nm, the nucleation temperature drops below the limit of homogeneous bulk nucleation. In **e**, the horizontal error bars denote variations in the dimension of the water droplets, and the vertical errors bar denotes experimental errors in the measurement of nucleation temperature.

nanodroplets confined in 150 nm pores and indicates water OH stretch peak at 3300 cm⁻¹. However, once the temperature of the system drops to −11 °C, the OH stretch peak becomes narrower and shifts to lower wavenumbers due to the stronger and less heterogeneous hydrogen bonding networks[44,45]. This narrowing and peak shift indicate water–ice phase transformation. We continued these experiments for other pore dimensions of 80, 40, 20, and 10 nm as shown in Fig. 3 and Supplementary Note 6. Similar to the previous metrology, we found size dependence of water–ice transformation temperature in these nanodroplets. As we have two pore dimensions in 5 and 2 nm membranes, ice OH stretch peak in smaller pores would be overshadowed by ice OH peaks in larger pores. To address this challenge, we only wetted one side of the membrane and at the same time, we introduced octane from the other side of the membrane to limit the water nanodroplets in the small pore. The results are shown in Fig. 3c, d for 5 and 2 nm pores and shows the OH stretch peak shift at much lower temperatures than large pores. This is in harmony with the findings of the electrical resistance metrology. The OH stretch bond in these few nanometer nanodroplets is similar to the larger nanodroplets, but FTIR metrology is not capable of distinguishing between cubic ice (Ic), and Isd. The size dependence of water–ice phase transformation of nanodroplets measured through two metrologies are compared in Fig. 3e. The results from both metrologies are in close agreement and indicate a sharp dependence of transformation temperature for droplets

smaller than 10 nm. We should emphasize that the observed size dependence of ice nucleation is not caused by the volume of water in the pores, see Supplementary Note 3. The volume of water inside the pores was measured using quartz crystal microbalance analysis and the results are tabulated in Supplementary Table 2, which shows that the membranes are substantially filled with water and the volume of liquid in all pore dimensions are in the same order.

To understand the length-scale dependence of heterogeneous phase transformation, we explored the role of length-scale on Gibbs energy barrier of water–ice phase change. The Gibbs energy barrier for heterogeneous nucleation ($\triangle G^*$) is written as[16,46]

$$\Delta G^* = \frac{16\pi\gamma_{IW}^3}{3(\rho\Delta\mu)^2}f(m,x) \qquad (1)$$

where $\gamma_{IW}$ denotes interfacial tension between water and ice, $\rho$ is the density of water, $\Delta\mu$ is the chemical potential difference between ice and water phases, and $f(m,x)$ is the surface function which depends on interfacial energies, $m$, and interface geometry, $x$ (Supplementary Note 7). The length-scale plays a role in $\rho$, $f(m,x)$ and $\Delta\mu$. For the nanodroplets that experience high pressure, the liquid density varies by ~1.5%[7]. As the nanodroplets are encapsulated by oil, the interface for heterogeneous ice nucleation (oil–water) is concave. The expression of surface

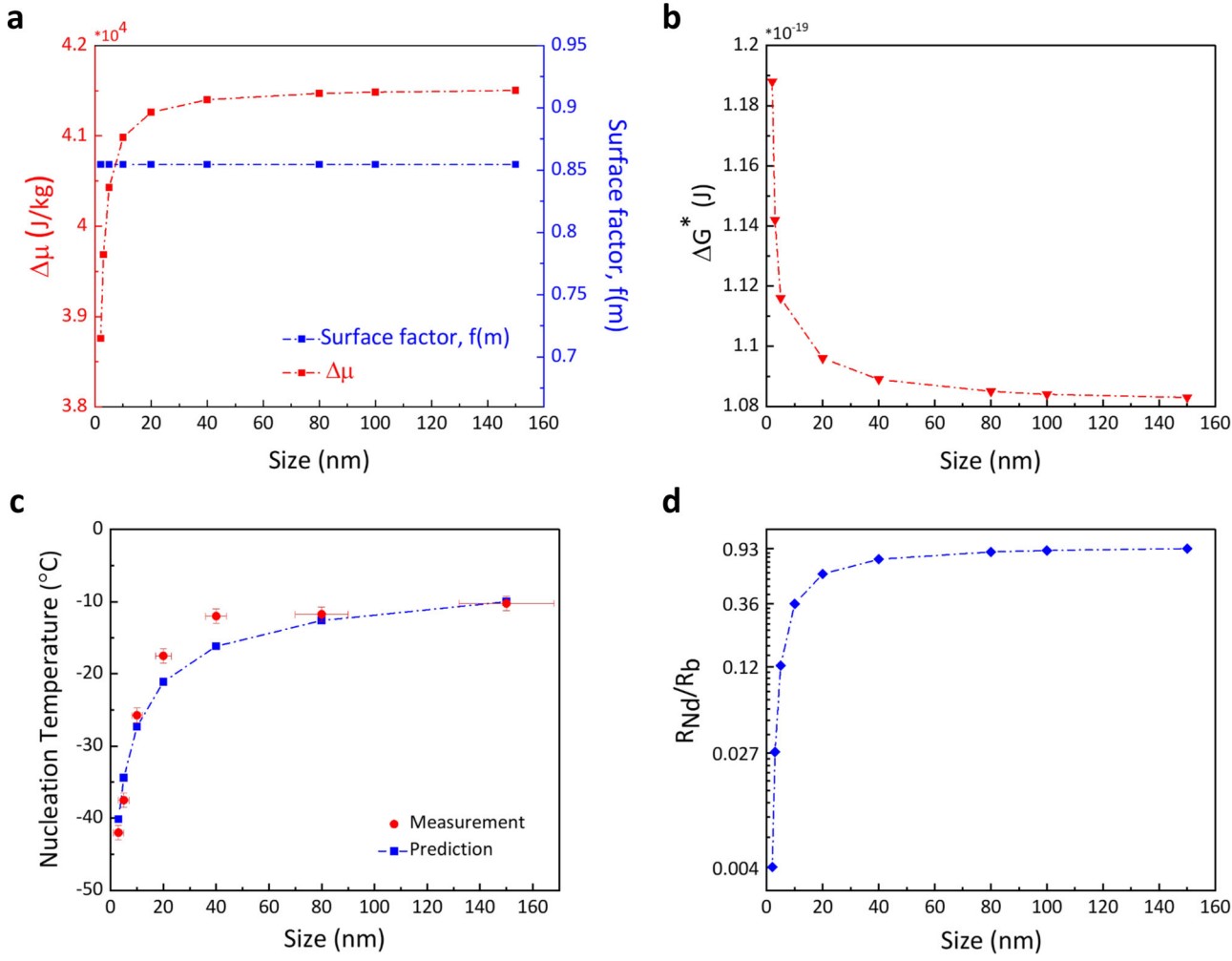

**Fig. 4 Gibbs energy barrier for freezing of nanoscopic water droplets. a** The dependence of chemical potential difference on length-scale is shown at $T = -41\,°C$. Due to the interfacial deformation, the role of interfacial curvature on surface function can be neglected and $f(m)$ is considered constant for all sizes. **b** The Gibbs energy barrier for water–ice phase change of nanodroplets is shown as a function of the diameter of nanodroplets at $T = -41\,°C$. **c** The predicted ice nucleation temperature by CNT for various drop sizes is compared with the averaged measured $T_N$ by two metrologies. The horizontal error bars denote variations in the dimension of the water droplets, and the vertical errors bar denotes experimental errors in the measurement of nucleation temperature. **d** The normalized ice nucleation rate of nanodroplets with respect to bulk water is shown as a function of length-scale at $T = 41\,°C$. The nucleation rate drops by more than two orders for a 2 nm nanodroplet.

function is given in Supplementary Note 7 and is plotted as a function of length-scale for both $-10$ and $-41\,°C$ in Supplementary Fig. 16. By considering the role of curvature, $f(m, x)$ function drops sharply for a few nanometer droplets leading to a drop of $\Delta G^*$ and consequently higher ice nucleation temperature for a few nanometer water droplets. Also through MD modeling, it has been shown that on solid concave interfaces, ice nucleation rate dramatically increases due to geometry effect on $\Delta G^*$[47]. Surprisingly, this is contradictory with the experimental findings above and points to a missing understanding here. As the oil–water interface is soft, Supplementary Fig. 17, once an ice nucleolus forms at this interface, the complete interfacial force balance needs to account for the unbalanced force of $\gamma_{IW}\sin\theta$, which leads to deformation of interface forming ripples at the oil–water interface[48]. In the case of stiff solid surfaces, this unbalanced force could be neglected due to the high modulus of solid surfaces. Thus, upon ice nucleation at the oil–water interface, the ripples form at the periphery of the water droplet and adjust the local curvatures of the interface. That is the oil–water interface forms combined convex and concave interfaces. Ice

nucleation occurs at both concave and convex coordinates. As the measured ice nucleation temperature is the average transformation temperature for the entire droplet, we may approximate zero curvature and determine $f(m, x)$ only as a function of $m$ as shown in Fig. 4a. Also, it should be noted that the effect of pore walls (to act as active sites) is negligible on ice nucleation here. Since the oil used in this study (octane) possesses a lower surface tension (21.6 mN/m) compared to water (72 mN/m), it is a safe assumption is that the inner surfaces of the pores are preferably wetted by the oil phase. Hence, water must be encompassed by the oil phase and should not have contact with pore walls and potential active sites. Moreover, while the $P\Delta v$ ($P$ is pressure difference at the oil–water interface and $\Delta v$ is the specific volume difference between water and ice) term showed a significant effect on ice nucleation, the surface factor did not, which suggests that pore walls do not play a role in nucleation here. This would have not been the case should water–wall interface be available for ice nucleation. Even if one considers that some pores have defects or water has contact with the pore wall, one observes the changes in both electrical resistance and FTIR spectrum when most of the

pores are frozen. In other words, freezing of a few pores does not show measurable changes in the results and significant changes occur when the majority of pores are frozen.

Theoretical analysis shows that if water has contact with pore walls and they act as nucleation sites, the effect of the surface factor becomes significant. To further demonstrate this fact, we filled membranes with water without the addition of oil, where water has contact with pores walls and nucleation starts from the wall active sites. In this case, we performed FTIR analysis on membranes with different pore dimensions to find $T_N$ as a function of size, and results are shown in Supplementary Note 8. It is observed that in this case, $T_N$ is higher even than the bulk nucleation temperature. For example, for a pore dimension of 2 nm filled just with water, through FTIR measurement, we have shown that ice forms at the temperature of ~0 °C. However, for the same pore dimension with the oil–water interface involved, ice forms at −42 °C. We should add that for fully water-filled pores, ice nucleation temperature increases with the pore dimension as discussed by Marcolli[49]. The difference in the trend compared to our work could be due to the extremely low pressure (negative pressure) of studied droplets here and its consequent effect on the chemical potential difference, see Supplementary Note 8.

The length-scale dependence of chemical potential manifests itself in the form of pressure and is written as[16,50]

$$\triangle\mu_{Nd}(T,P) = \triangle\mu_b(T,P_{atm}) + (P_{Nd} - P_{atm})(v_w - v_i) \quad (2)$$

in which $\triangle\mu_{Nd}(T,P)$ and $\triangle\mu_b(T,P_{atm})$ are the chemical potential difference between ice and water for a nanodroplet and bulk water, respectively, $(P_{Nd} - P_{atm})$ is pressure difference at the oil–water interface, and $v_w$ and $v_i$ are specific volumes of water and ice, respectively. We plotted the dependence of $\triangle\mu$ as a function of length-scale of nanodroplets in Fig. 4a. As shown, the chemical potential difference drops sharply for a few nanometer droplets due to high Laplace pressure. Having these dependencies, the Gibbs energy barrier for heterogeneous ice nucleation of nanodroplets is shown in Fig. 4b. As shown, in nanodroplets with dimensions of <10 nm, the Gibbs energy barrier is increased. Given the Gibbs energy barrier, we determined the theoretical ice nucleation temperature of nanodroplets based on the classical nucleation theory (CNT)[46], and plotted it in Fig. 4c. Details of ice nucleation temperature calculations are explained in Supplementary Note 9. We considered the same nucleation rate at $T_N$ for all the nanodroplets in these calculations. In addition, the average measured $T_N$ by two metrologies is included here. As shown, there is a good agreement between the measurement and predictions by CNT. The ratio of ice nucleation in nanodroplets compared to bulk value is written as[7]

$$\frac{R_{Nd}}{R_b} = \exp\left[-\frac{16\pi\gamma_{IW}^3 f(m)}{3k_BT\rho^2}\left(\frac{1}{(\Delta\mu_b + p\Delta v)^2} - \frac{1}{\Delta\mu_b^2}\right)\right] \quad (3)$$

This ratio as a function of length-scale is shown in Fig. 4d. Also, this ratio for the temperature of −10 °C is shown in Supplementary Fig. 24. At length scales smaller than 10 nm, the ice nucleation rate drops drastically compared to that in the bulk phase suggesting the suppression of ice formation in these nanodroplets.

## Discussion
We probed the ice nucleation of nanodroplets down to a 2 nm scale and found that interfacial deformation at soft interfaces and high pressure could significantly suppress ice nucleation rate and delay ice nucleation to temperatures even lower than homogeneous bulk nucleation temperature[51–53]. The pressures induced in these nanodroplets are up to ~500 bar and the only stable

phases of ice in this pressure range based on the phase-equilibrium data are Ih and Ic. However, it has been shown that ice formed by freezing of supercooled water forms Isd with a high degree of cubicity that anneals to stable hexagonal ice on the time scale of hours, and this transition is subject to the kinetics of recrystallization[54–57]. It has been discussed that nanoconfined water could show narrowing of OH peaks due to no free OH groups that are in the bulk phase. We did not observe this effect possibly due to the elongated ellipsoidal shape of nanodroplets here. The analysis of the finding through CNT suggests that the pressure in these nanodroplets is the governing factor in the suppression of ice nucleation and agreement between the finding and the CNT predictions supports the dominant role of pressure. It has been discussed that ice nucleation in water clusters containing 275 water molecules occurs at temperatures in the range of −183 to −158 °C[3] and some simulations have shown that ice nucleation in nanodroplets with a diameter of 2 nm can be suppressed down to −123 °C[58]. If we consider the case of homogeneous nucleation and assume the value of $f(m,x)$ equal to 1, the value of the Gibbs energy barrier could be increased by ~17% meaning that ice nucleation temperature (Fig. 4b, c) could be dropped by a few degrees. However, the more salient effect comes from the role of pressure in a spherical droplet. If one considers a hypothetical case of 2 nm spherical droplets in the air, the pressure in the droplets could be increased by approximately two times compared to the studied nanodroplets here. That is based on the decreased nucleation rate and extrapolation of the above findings; the solid–ice temperature could drop to −60 °C, which is still far from the predictions of −123 °C. It has been suggested that for nanoconfined water droplets in order of 1 nm, there are significant broken O–H bond[59] leading to an exotic state and for 2–3 nm confined droplets, ice-like nanocluster of water are formed[60] that could affect the solid–liquid phase-change characteristics. Furthermore, the phase-change transition could be completely non-monotonic[37]. The nature of the OH group in a few nanometer droplets and the finite number of molecules may be a key to addressing this difference. The findings provide an understanding of various natural phenomena and provide a route for the design of superior anti-icing biomimetics or smooth liquid-infused surfaces[21–23].

## Methods
**Confined nanodroplets.** The nanomembranes are acquired from InRedox Co and are cleaned with isopropyl alcohol (IPA) and plasma cleaner. The membrane was secured between the two reservoirs as shown in Supplementary Fig. 7. One side of the membrane was wetted with water and allowed for capillary force to drive the water into the pores. To ensure complete wetting of nanopores, the setup was placed in a sonicator for 5 min. Once the other side of the membrane is wet, it indicates the filling of the nanomembrane with water. We let the extra water on both sides of the membrane to evaporates for 30 min. Due to the wetting characteristics of water, water droplet adopts high negative pressure in the pore[35,61]. In the next step, both reservoirs were filled with octane to confine the water droplet in the nanopores. Octane wets the membrane faces and flows inside the pores. The final configuration of the nanodroplet in the pores is shown in Fig. 1. We examined the existence of water nanodroplets in the pores through electrical resistance measurements shown in Supplementary Fig. 5.

**Electrical resistance metrology.** Electrical resistance experiments were performed using a setup shown in Supplementary Note 4. In this setup, an aluminum plate with a hole in the middle was acquired and AAO membranes were mounted on top of the hole. The aluminum plate provides an isothermal condition in the membrane. The plate surface except the membrane area was coated with an insulator material to avoid any electrical shortcut through the plate. This plate was placed between two reservoirs and two electrodes placed on each side of the membrane such that the distances between electrodes were equal and the electrodes were as close as possible to the membrane to reduce oil resistance. The electrodes were connected to a high-resolution source meter (Keithley 2602B) to generate I–V curves at different temperatures. The I–V curves were produced in different temperatures using a Labview code. The setup was cooled down using TEC coolers connected to a power supply. The concept of this approach is clarified in Supplementary Note 4.

**FTIR metrology**. The membrane was placed in water and sonicated for 5 min. Then it was removed from water and we let the extra water on both sides of the membrane evaporate. A thin layer of octane is injected on the membrane in a parallel direction with the surface of the membrane to wash any water that may exist on the surface. This membrane was sandwiched between two glass coverslips and placed under Thermo Scientific Nicolet iS50 FTIR on a zinc selenide window to acquire FTIR spectra as a function of temperature. The details of this approach are explained in Supplementary Note 6.

## Data availability

The data generated in this study are included in the Supplementary information and are also available from the corresponding author upon request.

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

## Acknowledgements

We acknowledge the funding support by National Science Foundation (Grant No. 1804204) and ACS Petroleum Research Fund (Grant No. 59590) with Burtrand Lee as the program manager. The authors acknowledge the help of Siwakorn Sakunkaew-kasem in quartz crystal microbalance measurements.

## Author contributions

H.G. and A.H. conceived the idea. A.H. conducted the main experiments and M.M. and M.N., A.D., S.N. and Z. H. helped with the experiments. A.H. and H.G. wrote the manuscript with inputs from all the authors. All authors commented on the manuscript. J.B. and H.G. directed the research.

## Competing interests

H.G. is a co-founder of Elemental Coatings LLC specialized in the development of stress-localized durable icephobic surfaces. The other authors declare no competing interests.
