## [Peer Review File · Nature Communications]

Reviewers' comments:

Reviewer #1 (Remarks to the Author):

Hakimian et al. report a study on ice nucleation in water bubbles trapped in anodized aluminum oxide membranes with different pore sizes. From orthogonal electrical and FTIR measurements, the authors extract water nucleation temperatures T_N that decrease with decreasing pore size. Remarkably, in membranes with the smallest pores (2-4 nm), T_N is found to fall below the limit of homogeneous bulk nucleation, i.e., $T_N < -38$ C. The authors interpret their findings in terms of heterogeneous phase transformation where ice nucleation happens at the interface between water and oil (octane), the latter being applied to the outside of their membranes and believed to fully encapsulate the water phase in at least the majority of pores. The results are truly interesting and will certainly be of interest to the broad readership of Nature Communications. The work appears rigorous and the paper is very well written. Though I do have a few questions and some minor comments I'd like the authors to address before publication.

1) I am probably most puzzled by the electrical measurement and its interpretation. First of all, could the authors specify which electrode material they used? Second, are the I/V curves symmetric around zero or was only one polarity explored? Third, where and how do the authors think does the current flow (including what are the charge carriers)? A different way to ask this third question would be: where does most of the voltage drop occur? According to their interpretation, there is significant voltage drop across the bulk of the water droplet, the charge carriers likely being the ppm-level sodium and chloride ions. Are solvated sodium and chloride ions also held responsible for the charge transport inside the octane? How does the oil-water interfacial resistance compare to the bulk resistance of the water drop? And could there be parallel conduction inside the oil phase that dominates transport across the pores e.g. when ice has formed? Fourth, have the authors measured temperature-dependent I/V curves for octane-only filled membranes to rule out non-linearities in the octane resistivity?

2) Can the authors comment on the extent to which Span80 does or does not affect the oil-water interface?

3) If octane is believed to fully encapsulate water inside the pores, why does water remain inside the pores after all?

Minor comments:

- I feel discussion S3 is so important it should be in the main part of the paper. Although the FTIR results are a direct probe of water in the pores, the manuscript presents these results only after the electrical measurements have been discussed. The reader only confronted with the electrical measurements at first is left a bit wondering about the evidence for these water drops.
- The illustration of a carbon nanotube in Figure 1b to me is somewhat misleading.
- The caption to Figure S2 has a problem: a) should be anisotropic while b) should be isotropic.
- The parameters in “ $P \cdot \Delta \nu$ ” (page 6) are not introduced in the main text.

Reviewer #2 (Remarks to the Author):

Review of “Freezing of few nanometers of water droplets” by Hakimian et al.

The manuscript presents a methodologically sound approach for investigating the freezing properties of nano-sized pure water droplets in context of homogeneous and heterogeneous freezing (nucleation) mechanisms. The topic would be of fundamental interest to the atmospheric science community, amongst others. The experiments are interesting, the methods used are sound and the results are insightful. However, the interpretation leaves more to be desired, the novelty of the results is not clear to me and neither is the application of the results to a broad readership. From a fundamental standpoint, the results are highly interesting, but are likely to receive better attention in a more specialised journal like nanoscale or a nanoscience journal or even an atmospheric science journal.

The novelty of the study was not clear to me as it is presented. The results presented here are well known in the atmospheric science community, i.e. the suppression of freezing temperature of water in confined spaces and capillary environments. Reviews of such have been published a number of times and the authors have also cited one paper from 2019 that addresses this. Further, I would argue that the conclusions presented here are known, in particular from the atmospheric sciences field.

My overall remark scientifically, is that the interpretation of the results and the samples used is very unclear and potentially flawed (could be a misunderstanding in which clarifications are needed, see specific comments below) in some cases. The writing in the manuscript is not clear and requires careful editing. In my opinion this manuscript was not ready to submit for review, as the language, grammar and other simple aspects could be largely improved before submission. It is not a reviewer's role to contend with such issues. While I have tried my best to delineate between the

scientific review, sometimes the language confounded the issue of understanding the science presented herein.

In its present form, I would not recommend the manuscript for publication in Nature Communications, unless major revisions and the comments below are satisfactorily addressed. Furthermore, I would also recommend major revisions if this review was prepared for a more specialised journal. At this point, it would be impossible to give a recommendation for publishing before seeing a revised manuscript with the concerns below addressed. In addition to addressing the comments below, I would highly recommend that the authors carefully check their language for clarity not only with respect to grammar but also with respect to ice nucleation mechanisms and phase changes from liquid to ice vs. from vapour to liquid/ice. This appears to be confused a number of times in the first couple pages of the manuscript.

It would also help to have line numbers and page numbers to aid the reviewer for pointing out comments. This way, it is left upon the reviewer to decipher line numbers and reference points to the manuscript. Perhaps this is a comment relevant to the editorial team of Nature Communications and not the authors.

Page 1:

Line 1 and other locations in the manuscript: "Water/ice" should read "Water-ice" (just like other places in the manuscript, to be consistent, but this form is also more accurate since the authors are referring to a water to ice transformation).

Para. 2, line 4: should read "Ice nucleation in the atmosphere.."

The statement in this line as written is inherently flawed. In the atmosphere in the vapour environment alone, ice nucleation would not occur homogeneously. There would be a need for a liquid droplet (v. dilute solution droplet that could freeze homogeneously, or a concentrated solution droplet that could take up vapour, become dilute and freeze homogeneously) or a solution droplet for homogeneous freezing to occur. If the temperature is low enough, then the freezing point depression is overcome and the solution droplet (with dissolved ions) can freeze homogeneously. But ice nucleation occurring in the vapour environment is not called homogeneous and in any case this does not occur in the atmosphere, the Gibbs free energy for this is large (i.e. for

ice to form homogeneously out of the vapour phase without the aid of an aerosol particle (liquid or solid)) is practically impossible in the atmosphere.

Page 2

Line 1: The required number of water molecules to form crystalline ice is determined to be 273 molecules. But this should be made clear, are the authors talking about forming ice from the gas phase or forming ice from the liquid phase?

Line 3: the authors should specify here that they mean homogeneous freezing of nanoscopic water, because homogeneous nucleation of nanoscopic water implies that the water is nucleating from the gas phase, but what the authors mean is that ice is nucleating homogeneously from the liquid phase.

Para. 1 Line 3: not clear to me what type of infrastructures are being referred to here. Please clarify.

Para. 1. Line 5: Please define SLIPS and MAGSS (it is not sufficient that for an acronym, one needs to look up a reference).

Para 1. Line 6: "effect" should read "effects". Also, by confinement do the authors mean capillaries where water can condense and freeze within? This is not clear here.

Para 1. Lines 10-11: I suggest for better clarity and accuracy of language to change the sentence to "Hence, the study of nanodroplets in confinements is needed to accurately probe their phase transformation."

Para 2. Line 1: clarify if 2nm is radius or diameter of confinement, I think it is the diameter. Fig 1a is written as its own sentence, is this meant to be part of the sentence before or after?

Para 2. Line 3-4: is the dimension of 2 – 150 nm referring to diameter or radius?

Para 3. Line 2: Not clear what thickness means in this sentence. Does this refer to the length of the AAO membrane? Perhaps it would be beneficial to include a sketch in which the 50 – 60 micron thickness is indicated so a reader can appreciate the dimensionality of the AAO membranes used.

Page 3

The results section could be better organised into themes and paragraphs rather than one very long paragraph stretching over 2 pages.

Line 11: ..pores walls... should read "pore wall"

Line 27: The four-probe methods should be explained in the methods or the supplementary information and so should the parasitic resistances be explained somewhere. Even though the authors have cited literature, a brief explanation can be given in the methods or supplementary section.

Page 4

Line 2: I-V should be defined as current and voltage at its first use.

Page 5

Line 13-17: The authors state that they emphasize that the observed size dependence of ice nucleation is not caused by the volume of water in the pores. This was not clear to me why this is the case. Because the argument presented by comparing the total volume of water (Table S2) is then comparing total volume of water per cm² of the membrane. So even if the volume of water in the membrane (in total) as stated by the authors in the same order, what is key is the volume of the isolated droplet in the oil emulsion. Because for the isolated droplet to freeze, the ice germ needs to form within that volume. i.e., in their experiments, the authors are observing the freezing of nano-dimensioned water that is isolated in oil emulsions. As such what should be relevant for freezing is the volume of the isolated water droplet in oil, and not the entire volume of water in the membrane. Because for each isolated (in oil) droplet that froze, an ice germ has to form and for this, the ΔG barrier has to be overcome. In the case where oil is lining the pore walls the droplets are isolated from one another, so the total volume of water in the membrane should not matter. This is why the

ice nucleation temperature decreases substantially with size because it becomes increasingly energetically costly (difficult) to form an ice germ in smaller volume droplets.

The authors should consider also presenting the size of the critical ice germ for the given temperatures of nucleation to demonstrate if these ice germs can indeed be hosted in a 2 – 4 nm droplet at the observed freezing temperatures.

Line 26: It is not clear to me how the oil-water interface should be concave; it should be convex, and this is also shown in the figure in the supplementary material. If the authors say it should be concave, please clearly illustrate this in a schematic, that specifically addresses this, because this aspect is key in how the ΔG and the associated Laplace pressure is being assessed. And when this interface is concave, and the volume effect of nucleation is considered in the nano droplet, it is clear that ΔG should increase with decreasing dimensions of the capillary confinement such that the 2-4 nm experiments should demonstrate the lowest freezing temperature.

Page 6

Line 1: Similarly, this result is not surprising and not contradictory to what is expected. What was surprising to me is that the authors evaluated the ΔG to be decreasing with decreasing droplet dimension.

In this vein I would strongly urge the authors to carefully read Marcolli (2014), Marcolli (2017) and Marcolli (2020) in order to assess their interpretation of freezing of water in confinements and capillaries. The authors should reconsider the context of their discussion after reading the above papers. The concavity of the droplets should be addressed more clearly. A clear schematic or the pore structure is needed for the readers to better understand the absence of volume effect, if indeed that is true.

Line 15: Oil should coat the walls, not wet the walls

Line 23-32: This argument presented here is interesting. But I think the authors are mixing up three different issues here and each should be addressed more clearly. One is the fact that the authors state that significant changes in the results are only observed if majority of pores are frozen. This strikes me as a limit of detection issue with their methodology and nothing to do with classical nucleation theory, or the theory of ice forming mechanisms. This should be separated from the next two points.

Next comes the aspects of the freezing at higher temperatures in the absence of an oil surrounding environment for the water. This means that the water is in direct contact with the pore walls and the authors say that the pores contain active sites, which I agree is completely fathomable. The active sites can catalyse ice germ formation, reduce the ΔG required for ice nucleation and thus increase the freezing temperature. Nonetheless it strikes me as interesting that in the case where the total water content in the membrane is not isolated, i.e. the water volume can be considered as a whole, since the water droplets are not isolated by oil layers in between their volume is also now not as restricted as in the oil case. So how can the authors say it is only due to the active sites, and ignore again the volume effect here?

Also the freezing T being so high in the 2 – 4 nm pores suggests that the volume plays a role and that the volume of water to be taken into account is larger than that associated with the 2-4 nm dimension because at 273 K (the freezing temperature reported in the absence of oil) the size of an ice germ will be quite large. This calculation of the ice germ size as a function of freezing temperature must be presented in this work, in order to convince the reader that indeed water could freeze in the 2-4 nm confinements. The authors should also clarify that if there is no oil in the membrane, then is all the water volume considered continuous. i.e. is the water in one capillary in contact with another capillary. Based on how I understood the methods, the answer to this question is yes, and the oil experiments are the only ones that ensure the water is isolated into smaller volumes.

So, while I agree, that active sites play a role, it cannot be the only reason based on the information presented here in.

Page 7

Line 4: should read “..difference at the oil-water..”

Page 8

Line 1: “We did not observe..”

Line 3: should be nanodroplets

Line 5: should be clusters and replace “happens” with “occurs”

Line 6: “..temperatures in the range...”

Line 11: The example given here by the authors of one developing spherical 2nm droplets in air would not be possible because the Kelvin effect would be so high, that the droplets would have no practical chance of surviving. They would require unrealistically high super saturations with respect to liquid water in order to survive. As such the example that follows about dropping the ice nucleation temperature to 213 K is hypothetical at best. Unless the authors clarify in which application can spherical 2 nm droplets actually be formed in a stable manner to use in any application.

Figure 1: would be nice to have a legend indicating what is the oil layer here

Figure 2: I-V should be defined as current and voltage. Since the main indicator of phase change is resistance here, why not plot V on the y-axis and I on the x-axis since the slope of this curve then gives the resistance and then it is easy to assess, higher slope means higher resistance and when there is a jump in the slope that is where the phase change occurs. That would be more intuitive and physical for the plots shown in Figure 2.

Figure 3: For the smaller pore dimensions it seems that the FTIR metrology gives slightly lower temperatures for the phase transition. Have the authors considered what local heating effects the FTIR beam would have on the pore volume water. i.e. when the water droplets are so small and isolated, the FTIR beam could be providing some local heating therefore requiring further cooling in order to freeze. Could this be a potential reason for the slightly lower T with the FTIR metrology for the smaller pore size water freezing?

Supplementary Material

Figure S1. Can the authors harmonise the style on the figures? In some of the figures the labels of the legend have different colors and sizes and the labels on the figures as well. Also, it is not clear what the difference between “height” and “height sensor” is here. What does height refer to in these figures?

Figure S2: The authors often mention in the manuscript that the inner layer is the pore are concave, can this be schematically shown in this Figure. Also, can the dimension of 50 – 60 microns mentioned in the article also be depicted in this figure for clarity?

Figure S10: how are these results justified by CNT, without considering the total volume? It is the total active site density that matters, but that only matters if the active sites are all covered by a large volume of water. If the volume of water is really small, the probability it will cover an active site is also small. In this regard, the authors should demonstrate how an ice germ can form in a 2 nm pore at 273 K as demonstrated in this figure. In the caption the authors talk about curvature, I would expect because of water adhesive forces that the curvature is concave here. Can this be clarified as well?

References

Marcolli, C. (2014). "Deposition nucleation viewed as homogeneous or immersion freezing in pores and cavities." *Atmospheric Chemistry and Physics* 14(4): 2071-2104.

Marcolli, C. (2017). "Pre-activation of aerosol particles by ice preserved in pores." *Atmos. Chem. Phys.* 17(3): 1595-1622.

Marcolli, C. (2020). "Technical note: Fundamental aspects of ice nucleation via pore condensation and freezing including Laplace pressure and growth into macroscopic ice." *Atmos. Chem. Phys.* 20(5): 3209-3230.

Reviewer #3 (Remarks to the Author):

I agree that understanding the water/ice transformation is important in many different applications and that very few experiments can probe this transformation down to the nanoscale. Those that do often involve aerosols or large molecular clusters that are rapidly cooled in supersonic or free-jet expansions,¹⁻³ although recent work has also focused freezing in nanopores and nano-cages – both hydrophilic and hydrophobic.⁴⁻⁷ The use of the membranes to confine the lengthscale of the “droplets” is most similar to that of freezing in hydrophobic pores. Unfortunately, the current data are not compared to any of the recent literature that have studied freezing under confinement that use different measurement techniques. Nor does it try to relate the results to extensive literature regarding emulsion droplet freezing (see Refs 8,9, 10 and refs therein). Thus, the uniqueness and importance of the current work is not established, and the key point of the paper is not clear.

Some of the claims and assumptions made in the paper are neither demonstrated nor referenced.

1. The authors claim that the ice that forms is hexagonal even down to 2 nm without the definitive proof provided by a diffraction experiment. This result conflicts directly with the significant experimental and theoretical literature that finds that ice formed by freezing from supercooled water (homogeneously or heterogeneously) forms stacking disordered ice, ice I_{sd} , that only anneals to hexagonal ice on the time scale of hours.¹¹⁻¹³ Although recent Raman experiments¹⁴ have been able to distinguish I_{sd} from I_h , to this reviewer's knowledge the same is not true for IR experiments. The work of Bertie and Whalley¹⁵ states that hexagonal and cubic ice have IR spectra that are indistinguishable within the measurement limits. The reviewer is not aware of any papers that correct this view, and the authors do not provide evidence or even a reference. They simply state (without reference) that “The formed ice phase in the few nm droplets shows OH stretch peak of 3200 cm^{-1} rather than 3080 cm^{-1} (cubic) suggesting that all of the ice phases formed in the nanodroplets are I_h ”. This statement is also contradicted by the work of Amaya et al.¹⁶ and Amaya & Wyslouzil³ that showed droplets ~ 20 nm in diameter that froze near 225 K were highly cubic stacking disordered ice I (based on diffraction measurements) but their FTIR measurements look very similar to those shown here.
2. The FTIR experiments are not very well described in the main text or in the SI. There is no figure illustrating the setup. How was the temperature controlled? How well is the temperature known? How was water vapor prevented from condensing from the atmosphere? Where are the C-H stretch bands from the oil? Were they suppressed or subtracted? What was the resolution used to make the measurements?
3. The authors claim (without proof) that nucleation is heterogeneous and occurs at the oil water interface. To prove this requires experiments that demonstrate nucleation rates scale with available surface area or available volume. This paper does not characterize the water domains beyond stating that maximum radius is set by the tube diameter. After reading of the SI + the paper, it sounds more like the water is in tube-like structures (rather than spherical drops) with some constrained radius and variable/

unknown length. Without characterizing the domains, how can the current experiments prove what the nucleation mechanism is or whether it changes as a function of pore size?

There is a great deal of literature that discusses homogenous nucleation in water droplets surrounded by oil/surfactant mixtures.⁸⁻¹⁰ Most of these are not consistent with nucleation occurring at the oil-water interface, but rather with nucleation throughout the droplet. Why is this case different? How can that be proven?

4. The statement that the temperature for freezing 2 nm droplets drops below the “bulk homogeneous nucleation limit” is correct, but it is not surprising. Aerosol droplets in the same size range do not nucleate until the temperature is well below this limit as well.¹⁻³ It is not clear what is special here. Given that the claim is that nucleation at the interface will reduce the energy barrier to nucleation, shouldn't the comparison be to other droplets or water domains under comparable conditions and sizes rather than to bulk (micron sized)?
5. The authors chose octane to maximize the interfacial tension and to thereby ensure a high internal droplet pressure. They then add surfactant to the oil to improve electrical conductivity without discussing any possible effects this may have on interfacial tensions and, thus pressure. Nor do they discuss the effect that pressure can have on the predictions of classical theory although this has been discussed extensively in the literature especially by Espinoza and co-workers.¹⁷⁻¹⁸
6. The authors show data for water in contact with pores freezing at ~ -7 C for 20 nm pores, and water surrounded by oil in 150 nm pores freezing at ~ -10 C. What would be more inciteful would be experiments in the 150 nm pores in the absence of oil to tie the two types of experiments together.

This reviewer is not familiar with the electrical resistance methods used by the authors, but the results appear to be consistent with the FTIR measurements. The reviewer is familiar with FTIR measurements used to follow freezing and with nucleation theory.

Summary:

Overall I cannot recommend publishing this paper in Nature. The take home message/key point is not clear, there are a number statements and assumptions that are not proven or adequately discussed, and the literature includes papers that are not related to the current work (pore condensation papers) while ignoring much of the literature that is relevant. Some of the more important papers are cited as references in this review.

References:

1. Huang, J. F.; Bartell, L. S. Kinetics of homogeneous nucleation in the freezing of large water clusters. *Journal of Physical Chemistry* **1995**, *99*, 3924-3931.
2. Manka, A.; Pathak, H.; Tanimura, S.; Woelk, J.; Strey, R.; Wyslouzil, B. E. Freezing water in no-man's land. *Physical Chemistry Chemical Physics* **2012**, *14*, 4505-4516.
3. Amaya, A. J.; Wyslouzil, B. E. Ice nucleation rates near 225 K. *Journal of Chemical Physics* **2018**, *148* Art No. 084501.
4. Emily B. Moore, James T. Allen, Valeria Molinero, Liquid-Ice Coexistence below the Melting Temperature for Water Confined in Hydrophilic and Hydrophobic Nanopores, *J. Phys. Chem. C* **2012**, *116*, *13*, 7507-7514
5. Jahwar Jelassi, Hessel L. Castricum, Marie-Claire Bellissent-Funel, John Dore, J. Beau, W. Webberd and Rachida Sridi-Dorbez, Studies of water and ice in hydrophilic and hydrophobic mesoporous silicas: pore characterisation and phase transformations, *Phys. Chem. Chem. Phys.*, **2010**, *12*, 2838-2849
6. Kunimitsu Morishige, Influence of Pore Wall Hydrophobicity on Freezing and Melting of Confined Water, *Journal of Physical Chemistry C* **2018** *122* (9), 5013-5019
7. Simone Mascotto, Wolfhard Janke, and Rustem Valiullin, Ice Nucleation in Periodic Arrays of Spherical Nanocages, *J. Phys. Chem. C* **2017**, *121*, 23788-23792
8. Riechers, B.; Wittbracht, F.; Huetten, A.; Koop, T. The homogeneous ice nucleation rate of water droplets produced in a microfluidic device and the role of temperature uncertainty. *Physical Chemistry Chemical Physics* **2013**, *15*, 5873-5887.
9. Atkinson, J. D.; Murray, B. J.; O'Sullivan, D. Rate of Homogenous Nucleation of Ice in Supercooled Water. *Journal of Physical Chemistry A* **2016**, *120*, 6513-6520.
10. Mark D. Tarn, Sebastien N. F. Sikora, Grace C. E. Porter, Daniel O'Sullivan, Mike Adams, Thomas F. Whale, Alexander D. Harrison, Jesús Vergara-Temprado, Theodore W. Wilson, Jung-uk Shim, Benjamin J. Murray, The study of atmospheric ice-nucleating particles via microfluidically generated droplets, *Microfluidics and Nanofluidics* (2018) 22:52
11. Murray, B. J.; Knopf, D. A.; Bertram, A. K. The formation of cubic ice under conditions relevant to Earth's atmosphere. *Nature* **2005**, *434*, 202-205.
12. Benjamin J. Murray and Allan K. Bertram*, Formation and stability of cubic ice in water droplets, *Phys. Chem. Chem. Phys.*, **2006**, *8*, 186-192
13. Tamsin L. Malkin, Benjamin J. Murray, Christoph G. Salzmann, Valeria Molinero, Steven J. Pickering and Thomas F. Whale, *Stacking disorder in ice I*, *Phys.Chem.Chem.Phys.*, **2015**, *17*, 60
14. Milva Celli, Lorenzo Ulivi, and Leonardo del Rosso, Raman Investigation of the Ice Ic-Ice Ih Transformation, *J. Phys. Chem. C* **2020**, *124*, 17135-17140
15. J. E. Bertie, and E. Whalley, Infrared Spectra of Ices Ih and Ic in the Range 4000 to 350 cm^{-1} , *J. Chem. Phys.* **40**, 1637 (1964)
16. Amaya, A. J.; Pathak, H.; Modak, V. P.; Laksmono, H.; Loh, N. D.; Sellberg, J. A.; Sierra, R. G.; McQueen, T. A.; Hayes, M. J.; Williams, G. J.; Messerschmidt, M.; Boutet, S.; Bogan, M. J.; Nilsson, A.; Stan, C. A.; Wyslouzil, B. E. How Cubic Can Ice Be? *Journal of Physical Chemistry Letters* **2017**, *8*, 3216-3222.

17. Espinosa, J. R.; Zaragoza, A.; Rosales-Pelaez, P.; Navarro, C.; Valeriani, C.; Vega, C.; Sanz, E. Interfacial Free Energy as the Key to the Pressure-Induced Deceleration of Ice Nucleation. *Physical Review Letters* **2016**, *117* Art No.135702.
18. Jorge R. Espinosa, Carlos Vega, and Eduardo Sanz, Homogeneous Ice Nucleation Rate in Water Droplets, *Phys. Chem. C* 2018, *122*, 22892–22896

Revision Report

Reviewer #1:

“Hakimian et al. report a study on ice nucleation in water bubbles trapped in anodized aluminum oxide membranes with different pore sizes. From orthogonal electrical and FTIR measurements, the authors extract water nucleation temperatures T_N that decrease with decreasing pore size. Remarkably, in membranes with the smallest pores (2-4 nm), T_N is found to fall below the limit of homogeneous bulk nucleation, i.e., $T_N < -38$ C. The authors interpret their findings in terms of heterogeneous phase transformation where ice nucleation happens at the interface between water and oil (octane), the latter being applied to the outside of their membranes and believed to fully encapsulate the water phase in at least the majority of pores. The results are truly interesting and will certainly be of interest to the broad readership of Nature Communications. The work appears rigorous and the paper is very well written. Though I do have a few questions and some minor comments I'd like the authors to address before publication.”

Comments:

1) *“I am probably most puzzled by the electrical measurement and its interpretation. First of all, could the authors specify which electrode material they used? Second, are the I/V curves symmetric around zero or was only one polarity explored? Third, where and how do the authors think does the current flow (including what are the charge carriers?)? A different way to ask this third question would be: where does most of the voltage drop occur? According to their interpretation, there is significant voltage drop across the bulk of the water droplet, the charge carriers likely being the ppm-level sodium and chloride ions. Are solvated sodium and chloride ions also held responsible for the charge transport inside the octane? How does the oil-water interfacial resistance compare to the bulk resistance of the water drop? And could there be parallel conduction inside the oil phase that dominates transport across the pores e.g. when ice has formed? Fourth, have the authors measured temperature-dependent I/V curves for octane-only filled membranes to rule out non-linearities in the octane resistivity?”*

Author reply: We used copper electrodes for 4-point probing method in our experiments. This method ensures irrelevant electrode and contact resistances (*Kazani, I. et al. About the collinear four-point probe technique's inability to measure the resistivity of anisotropic electroconductive fabrics. Text. Res. J. 83, 1587–1593 (2013)*). One of the advantages of this method is the elimination of contact and wire resistances from the measurement.

A 4-point probe consists of four electrical probes in a line, with equal spacing between each of the probes. Based on this method, a high impedance current source is used to supply current through the outer two probes; a voltmeter measures the voltage across the inner two probes (See figures below). Due to the high impedance of voltmeter, no current flows through the inner electrodes. In this case, the voltage drop is measured between two inner electrodes where wire resistances (R_{w2} and R_{w3}) and contact resistances (R_{c2} and R_{c3}) do not contribute in voltage measurement and it is just the resistance of sample (R_{s2}) which leads to the voltage drop (ΔV).

Figure 1. 4-point probing method concept

We explored one polarity where we applied current and measured the voltage (4-point probing method procedure) and I/V curves are not symmetric around zero. In fact, what is important here is non-linearities in the resistance and by this method, we can observe those non-linearities and jump in the resistance due to solid-liquid phase change.

Based on the 4-point probing method, the voltage is measured between two inner electrodes and these two electrodes are pretty close to the membrane, almost attached to the membrane. As a result, almost all of the voltage drop here comes from the membrane and the liquid inside the membrane and the changes observed in the resistance is due to the characteristic changes in the liquid inside the membrane. The liquid phase contains water and octane. Octane conductivity is due to the existence of a tiny amount of Span80 and conductivity of water is due to the sodium and chloride ions. Addition of sodium and chloride ions increases the conductivity of water due to ion conductivity. The most important feature is that, as water freezes, ions cannot move and this increases ice resistance compared to water by three orders of magnitude (the order changed from $k\Omega$ to $M\Omega$ upon freezing in our measurement setup). On the other hand, when there is no ion inside the water, the resistance of ice and water is in the same order of magnitude. It should be noted that these ions are not soluble in octane, as octane is completely non-polar, and after adding NaCl to octane it precipitates and octane resistivity, which is in order of Giga ohm ($G\Omega$), does not change. Thus, sodium and chloride ions cannot be responsible for the charge transport inside the octane. Also, HLB (Hydrophilic–lipophilic balance) value for span 80 is 4.3 which means that span 80 is oil soluble and water-insoluble.

Regarding the interfacial resistance, what is important here is the non-linearities in the resistance and those non-linearities occur due to the phase change in water or octane. As the freezing point of octane is -57°C , these non-linearities is attributed to the phase change in water. Also, there can be parallel conduction inside the oil phase. However, as explained before, what is important here is the non-linearities in the resistance and those non-linearities occur due to the phase change in water.

We measured temperature-dependent I/V curves for octane-only filled membranes for 80 nm

membrane and results are shown in the figure below. From this figure, we cannot observe any non-linearity down to -14°C . However, we observed non-linearity in resistance in water-filled 80 nm membranes around -12°C .

Figure 2. I-V curves measured across the membrane when there is no water inside the pores in the membrane with pore diameter of 80 nm. There is no nonlinear jump in electrical resistance down to -14°C .

Action: 4point-probing method is explained and illustrated in the SI section.

The above plot is added to the supplementary information.

2) “Can the authors comment on the extent to which Span80 does or does not affect the oil-water interface?”

Author reply: HLB (Hydrophilic–lipophilic balance) value for span 80 is 4.3 which means that span 80 is oil soluble and water-insoluble (*Y. Lo, Relationships between the hydrophilic–lipophilic balance values of pharmaceutical excipients and their multidrug resistance modulating effect in Caco-2 cells and rat intestines, Journal of Controlled Release 90, 37-48 (2003)*). Furthermore, we used pendant drop method to access the role of Span80 concentration on octane-water interfacial tension. In these experiments, a water droplet is introduced in an octane medium with different concentrations of Span80 as shown below. The Bond number (Bo) is defined as

$$Bo = \frac{\Delta\rho g R_0^2}{\gamma_{ow}}$$

Where $\Delta\rho$ denotes the density difference between water and octane, g is the gravitational

acceleration and R_0 is the radius of curvature at the drop apex, **Figure 3**. The differential form of Young-Laplace equation in terms of arc length (s) is written as

$$\frac{d\phi}{d\bar{s}} = 2 - Bo \bar{z} - \frac{\sin \phi}{\bar{r}}$$

$$\frac{d\bar{r}}{d\bar{s}} = \cos \phi$$

$$\frac{d\bar{z}}{d\bar{s}} = \sin \phi$$

where the bar indicates dimensionless values scaled by R_0 . The boundary conditions are

$$@ s = 0: \bar{r} = 0; \bar{z} = 0; \phi = 0$$

The input to this system of equations is Bo or specifically γ_{ow} .

Figure 3. Schematic of pendant drop experiments.

Once this value is given, one can solve this system of equations numerically. We determined numerically the shape of the droplet for a range of surface tension values and compared the calculated droplet shape with the one measured, **Figure 4**. For each considered value of surface tension, we determined the coefficient of determination (R^2) between the calculated shape and measured shape. The surface tension with the highest value of R^2 gives us γ_{ow} through pendant droplet method, **Table 1**. As a control experiment, we compared the measured interfacial tension for octane-water with the reported value in the literature which shows less than 5% error.

Figure 4. The measured shape of water droplet suspended in octane are compared with the calculated shape to determine the surface tension of octane-water interface as a function of Span80 concentration. Note that even up to the concentration of 2500 ppm, the changes in surface tension is less than 5%.

Table 1: Surface tension of oil-water interface as a function of Span80 concentration

Concentration of Span 80	γ_{ow} in literature (mNm^{-1})	Measured γ_{ow} (mNm^{-1})
0 ppm	51	50
10 ppm		49
50 ppm		51
250 ppm		50
4500 ppm		48

Action: The new experiments are added to the SI of the manuscript.

3) “If octane is believed to fully encapsulate water inside the pores, why does water remain inside the pores after all?”

Author reply: The surface tension of octane is much lower than water. Thus, the octane wets the walls to make the system stable. On the other hand, there is no flow or pressure driven displacement in our system that leads to the removal of water from the pores [S. Liu et al., *Flow patterns of oil-water*

two-phase flow during pressure-driven process in nanoscale fluidic chips, Microfluidics and Nanofluidics (2018) 22:39] and all experiments are conducted on an optical table to make the setup completely stable and prevent any movement that may cause the water droplets to flow out of the pores. In addition, due to higher density of water, buoyancy force cannot force the water out of the pores. The results obtained from FTIR in the main manuscript prove the existence of water inside the pores.

Minor comments:

- *I feel discussion S3 is so important it should be in the main part of the paper. Although the FTIR results are a direct probe of water in the pores, the manuscript presents these results only after the electrical measurements have been discussed. The reader only confronted with the electrical measurements at first is left a bit wondering about the evidence for these water drops.*
- *The illustration of a carbon nanotube in Figure 1b to me is somewhat misleading.*
- *The caption to Figure S2 has a problem: a) should be anisotropic while b) should be isotropic.*

Author reply: Thank you. We added a sentence to provide more info on the existence of the water. However, due to space limitations in Fig. 2, we could not move the figure to the main manuscript. Also, the flow of the manuscript would have been completely changed if we were to bring the FTIR section before resistivity metrology. We hope this is reasonable to the reviewer.

Figure 1b is changed to be more illustrative. Also, as the water droplet is encapsulated with oil, the solid surface only plays a role in the curvature of liquid-liquid interface. The shown CNT is representative of any cylindrical nanometer pores.

The caption of Figure S2 is corrected.

Action: Changes are made in the main manuscript.

- *The parameters in “ $P\Delta\nu$ ” (page 6) are not introduced in the main text.*

Author reply: The parameter $P\Delta\nu$ is introduced in the main text.

Action: The changes are applied in the main text on page 6.

We again thank the reviewer for the thoughtful comments.

Reviewer #2:

“The manuscript presents a methodologically sound approach for investigating the freezing properties of nano-sized pure water droplets in context of homogeneous and heterogeneous freezing (nucleation) mechanisms. The topic would be of fundamental interest to the atmospheric science community, amongst others. The experiments are interesting, the methods used are sound and the results are insightful. However, the interpretation leaves more to be desired, the novelty of the results is not clear to me and neither is the application of the results to a broad readership. From a fundamental standpoint, the results are highly interesting, but are likely to receive better attention in a more specialized journal like nanoscale or a nanoscience journal or even an atmospheric science journal.

The novelty of the study was not clear to me as it is presented. The results presented here are well known in the atmospheric science community, i.e. the suppression of freezing temperature of water in confined spaces and capillary environments. Reviews of such have been published a number of times and the authors have also cited one paper from 2019 that addresses this. Further, I would argue that the conclusions presented here are known, in particular from the atmospheric sciences field.

My overall remark scientifically, is that the interpretation of the results and the samples used is very unclear and potentially flawed (could be a misunderstanding in which clarifications are needed, see specific comments below) in some cases. The writing in the manuscript is not clear and requires careful editing. In my opinion this manuscript was not ready to submit for review, as the language, grammar and other simple aspects could be largely improved before submission. It is not a reviewer’s role to contend with such issues. While I have tried my best to delineate between the scientific review, sometimes the language confounded the issue of understanding the science presented herein.

In its present form, I would not recommend the manuscript for publication in Nature Communications, unless major revisions and the comments below are satisfactorily addressed. Furthermore, I would also recommend major revisions if this review was prepared for a more specialized journal. At this point, it would be impossible to give a recommendation for publishing before seeing a revised manuscript with the concerns below addressed. In addition to addressing the comments below, I would highly recommend that the authors carefully check their language for clarity not only with respect to grammar but also with respect to ice nucleation mechanisms and phase changes from liquid to ice vs. from vapor to liquid/ice. This appears to be confused a number of times in the first couple pages of the manuscript.

It would also help to have line numbers and page numbers to aid the reviewer for pointing out comments. This way, it is left upon the reviewer to decipher line numbers and reference points to the manuscript. Perhaps this is a comment relevant to the editorial team of Nature Communications and not the authors.”

Author reply: Thanks for the comments. We have elaborated all these aspects in the below section. Just as a synopsis, the novelty of this work is that (1) for just water-filled pores, as one moves to few nm scales concave surfaces, the ice nucleation temperature increases (0°C for 2 nm water

droplet). However, once oil encapsulates these droplets, the trend completely changes, and ice nucleation temperature decreases (-41°C for 2 nm water droplet encapsulated with oil). Note that for both of these droplets, the ice nucleating surface is concave. Also, the Laplace pressure effect by itself (i.e. through its role on chemical potential) cannot justify this contrast in the findings. Furthermore, (2) contrary to nucleation on curved stiff solid surfaces, ice formation on soft interfaces (omnipresent in nature) could deform the interface leading to suppression of ice nucleation. We have provided even further experiments to support point (2).

Comments:

1) *“Line 1 and other locations in the manuscript: “Water/ice” should read “Water-ice” (just like other places in the manuscript, to be consistent, but this form is also more accurate since the authors are referring to a water to ice transformation).”*

Author reply: “Water/ice” is changed to “Water-ice”.

Action: Changes are made in the manuscript.

2) *“Para. 2, line 4: should read “Ice nucleation in the atmosphere..” The statement in this line as written is inherently flawed. In the atmosphere in the vapor environment alone, ice nucleation would not occur homogeneously. There would be a need for a liquid droplet (v. dilute solution droplet that could freeze homogeneously, or a concentrated solution droplet that could take up vapor, become dilute and freeze homogeneously) or a solution droplet for homogeneous freezing to occur. If the temperature is low enough, then the freezing point depression is overcome and the solution droplet (with dissolved ions) can freeze homogeneously. But ice nucleation occurring in the vapor environment is not called homogeneous and in any case this does not occur in the atmosphere, the Gibbs free energy for this is large (i.e. for ice to form homogeneously out of the vapor phase without the aid of an aerosol particle (liquid or solid)) is practically impossible in the atmosphere.*

Author reply: The sentence is changed to *“Ice nucleation in atmosphere could occur either on particles (heterogeneous) or in a liquid droplet (or a dilute solution droplet) surrounded by a vapor environment (homogeneous)”*.

Action: The changes are made in the main manuscript.

3) *“Page 2, Line 1: The required number of water molecules to form crystalline ice is determined to be 273 molecules. But this should be made clear, are the authors talking about forming ice from the gas phase or forming ice from the liquid phase?”*

Author reply: We mean the formation of ice from the liquid phase. In this manuscript, we studied freezing (phase change from liquid to solid) not desublimation (phase change from vapor to solid).

Action: This is clarified in the manuscript.

4) *“Page 2, Line 3: the authors should specify here that they mean homogeneous freezing of nanoscopic water, because homogeneous nucleation of nanoscopic water implies that the water is*

nucleating from the gas phase, but what the authors mean is that ice is nucleating homogeneously from the liquid phase.”

Author reply: We mean the homogeneous nucleation of ice from the liquid phase.

Action: The changes are made in the main manuscript.

5) *“Page 2, Para. 1 Line 3: not clear to me what type of infrastructures are being referred to here. Please clarify.”*

Author reply: The infrastructures include transportation systems, power systems, and energy systems. For example, icing in aircrafts increases drag and leads to loss of lift force and potential catastrophic events. Icing in electricity transmission systems can lead to the collapse of poles and towers and the rupture of conductors. Icing in cooling systems significantly drops the heat transfer rate leading to their inefficient operation.

Action: References of 14-16 have provided a detailed impact of icing on infrastructures.

6) *“Para. 1. Line 5: Please define SLIPS and MAGSS (it is not sufficient that for an acronym, one needs to look up a reference).”*

Author reply: SLIPS stands for slippery liquid infused porous surfaces and MAGSS stands for magnetic slippery surfaces.

Action: These names are defined in the main manuscript.

7) *“Para 1. Line 6: “effect” should read “effects”. Also, by confinement do the authors mean capillaries where water can condense and freeze within? This is not clear here.”*

Author reply: Yes, confinement can be a capillary, pore or channel with small dimensions.

Action: The change for the “effect” is made in the main manuscript.

8) *“Para 1. Lines 10-11: I suggest for better clarify and accuracy of language to change the sentence to “Hence, the study of nanodroplets in confinements is needed to accurately probe their phase transformation.”*

Author reply: The sentence is changed with the proposed sentence.

Action: The change is made in the main manuscript.

9) *“Para 2. Line 1: clarify if 2nm is radius or diameter of confinement, I think it is the diameter. Fig 1a is written as its own sentence, is this meant to be part of the sentence before or after?”*

Author reply: 2 nm is the diameter of droplet. Fig 1a is a part of the sentence before that.

Action: This is clarified in the manuscript.

10) *“Para 2. Line 3-4: is the dimension of 2 – 150 nm referring to diameter or radius?”*

Author reply: The dimension of 2-150 nm refers to the diameter of droplets.

Action: This is clarified in the manuscript.

11) “Para 3. Line 2: Not clear what thickness means in this sentence. Does this refer to the length of the AAO membrane? Perhaps it would be beneficial to include a sketch in which the 50 – 60 micron thickness is indicated so a reader can appreciate the dimensionality of the AAO membranes used.”

Author reply: We used membranes with different pore sizes ranging from 2 to 150 nm. However, all of these membranes have the same diameter and thickness. The diameter of membranes we used is 1 cm and their thickness is 50-60 microns (See figure below). In other words, the thickness of membrane is equal to the length of pores.

Action: These figures are added to Figure S2.

Figure 5. Schematic of top and side view of AAO membranes. In this photo (side view) thickness of membrane is clear.

Page3: The results section could be better organized into themes and paragraphs rather than one very long paragraph stretching over 2 pages.

Action: We re-organized the results in a few paragraphs.

12) “Page 3, Line 11: ..pores walls... should read “pore wall”.”

Author reply: “Pores walls” is changed to “pore wall”.

Action: The change is made in the main manuscript.

13) “Page 3, Line 27: The four-probe methods should be explained in the methods or the supplementary information and so should the parasitic resistances be explained somewhere. Even

though the authors have cited literature, a brief explanation can be given in the methods or supplementary section.”

Author reply: A 4-point probe consists of four electrical probes in a line, with equal spacing between each of the probes. Based on this method, A high impedance current source is used to supply current through the outer two probes; a voltmeter measures the voltage across the inner two probes (See figures below). Due to the high impedance of voltmeter, no current flows through the inner electrodes. In this case, the voltage drop is measured between two inner electrodes where wire resistances (R_{W2} and R_{W3}) and contact resistances (R_{c2} and R_{c3}) do not contribute in voltage measurement and it is just the resistance of sample (R_{s2}) which leads to the voltage drop (ΔV).

This method ensures irrelevant electrode and contact resistances (Kazani, I. et al. About the collinear four-point probe technique’s inability to measure the resistivity of anisotropic electroconductive fabrics. Text. Res. J. 83, 1587–1593 (2013)). One of the advantages of this method is the elimination of contact and wire resistances from our measurement.

Action: 4point-probing is explained and illustrated in SI.

Figure 6. 4-point probing method concept

14) “Page 4, Line 2: I-V should be defined as current and voltage at its first use.”

Author reply: I-V is defined as current and voltage at its first use.

Action: The change is made in the main manuscript.

15) “Page 4, Line 2: I-V should be defined as current and voltage at its first use.”

Author reply: I-V is defined as current and voltage at its first use.

Action: The change is made in the main manuscript.

16) “Line 13-17: The authors state that they emphasize that the observed size dependence of ice nucleation is not caused by the volume of water in the pores. This was not clear to me why this is the case. Because the argument presented by comparing the total volume of water (Table S2) is then comparing total volume of water per cm² of the membrane. So even if the volume of water in the membrane (in total) as stated by the authors in the same order, what is key is the volume of the isolated droplet in the oil emulsion. Because for the isolated droplet to freeze, the ice germ needs to form within that volume. i.e., in their experiments, the authors are observing the freezing of nano-dimensioned water that is isolated in oil emulsions. As such what should be relevant for freezing is the volume of the isolated water droplet in oil, and not the entire volume of water in the membrane. Because for each isolated (in oil) droplet that froze, an ice germ has to form and for this, the ΔG barrier has to be overcome. In the case where oil is lining the pore walls the droplets are isolated from one another, so the total volume of water in the membrane should not matter. This is why the ice nucleation temperature decreases substantially with size because it becomes increasingly energetically costly (difficult) to form an ice germ in smaller volume droplets.

The authors should consider also presenting the size of the critical ice germ for the given temperatures of nucleation to demonstrate if these ice germs can indeed be hosted in a 2 – 4 nm droplet at the observed freezing temperatures.”

Author reply: Thank you for this insightful comment that helped us to clarify the droplet volume effect. We agree with the reviewer that the key is the volume of isolated droplets in the pore. The droplets in these pores are elongated ellipsoid droplets with smaller diameter in the range of few nm while the other diameter in tens of micrometers. The approximate volume of an isolated water droplet in the pores are given in the table below:

Table 2. Volume of water inside each single pore of membranes with different pore diameter

Pore Diameter (nm)	Volume of water inside each pore (pL)
150	8.83×10^{-4}
80	2.51×10^{-4}
40	6.28×10^{-5}
20	1.57×10^{-5}
10	3.92×10^{-6}
5	1.17×10^{-7}
2-4	4.24×10^{-8}

We compared the volume of these droplets with the volume of water droplets reported in **Li, T., Donadio, D. & Galli, G. Ice nucleation at the nanoscale probes no man’s land of water. Nat. Commun. 4, 1887 (2013)** below. As shown, for water droplets larger than 6.1 nm, the ice nucleation rate is similar to bulk water. The volume of 6.1 nm droplet is 9.5×10^{-10} pL. All of the droplets studied in this work have volumes at least 2 orders of magnitude higher than this limit. Thus, the volume effect on ice nucleation rate is insignificant in this study.

Figure 1 | Size dependence of ice nucleation rates in the mW water droplets at 230 K. The solid black squares denote the calculated ice

On the size of critical ice germ, it is already presented in Table S3 and it can be observed that ice germ can be hosted in a 2-4 nm droplet at the measured freezing temperatures.

Action: The discussion on the volume effect is added to the SI.

17) “Line 26: It is not clear to me how the oil-water interface should be concave; it should be convex, and this is also shown in the figure in the supplementary material. If the authors say it should be concave, please clearly illustrate this in a schematic, that specifically addresses this, because this aspect is key in how the ΔG and the associated Laplace pressure is being assessed. And when this interface is concave, and the volume effect of nucleation is considered in the nano droplet, it is clear that ΔG should increase with decreasing dimensions of the capillary confinement such that the 2-4 nm experiments should demonstrate the lowest freezing temperature.”

Author reply: Due to the fact that ice nucleates inside the water not oil, the oil is a concave substrate for ice nucleation as shown below. Also, for concave interface, ΔG decreases as the size of droplet decreases. (i.e. $f(m,x)$ becomes smaller for smaller droplets.), **Fig. S11.**

Figure 7. Schematic of water nano droplet encapsulated in oil and the formation of ice embryo inside water.

Action: We added these figures to the SI.

18) “Page 6, Line 1: Similarly, this result is not surprising and not contradictory to what is expected. What was surprising to me is that the authors evaluated the DG to be decreasing with decreasing droplet dimension.

In this vain I would strongly urge the authors to carefully read Marcolli (2014), Marcolli (2017) and Marcolli (2020) in order to assess their interpretation of freezing of water in confinements and capillaries. The authors should reconsider the context of their discussion after reading the above papers. The concavity of the droplets should be addressed more clearly. A clear schematic or the pore structure is needed for the readers to better understand the absence of volume effect, if indeed that is true.”

Author reply: First, we discuss the referred literature and after we address the raised points:

*Marcolli, C. (2014). "Deposition nucleation viewed as homogeneous or immersion freezing in pores and cavities." Atmospheric Chemistry and Physics 14(4): 2071-2104.

Author reply: This article studied deposition nucleation which is pore condensation and freezing inside the pores (PCF). The condensation of water occurs in pores in humid air below water saturation by the inverse Kelvin effect, followed by homogeneous or heterogeneous freezing. In this article, the authors concluded that nucleation in immersion mode is less probable than homogenous nucleation, due to the need for at least one active site in a water-filled pore. Thus, by considering the homogeneous nucleation surface factor, $f(m, x)$ equal to 1, Gibbs energy barrier (ΔG^*) is written as

$$\Delta G^* = \frac{16\pi\gamma_{IW}^3}{3(\rho\Delta\mu)^2} \quad (4)$$

In our study, the water droplets are encapsulated with oil and no vapor condensation occurs. In the case of only water droplets in the pores, as shown in FTIR metrology, experiments are conducted in an environment with low humidity (purging N₂ gas) to avoid frost formation and water may not condense inside the pores (see Figure below).

Figure 8. Schematic of FTIR analysis setup that is used.

* Marcolli, C. (2017). "Pre-activation of aerosol particles by ice preserved in pores." *Atmos. Chem. Phys.* 17(3): 1595-1622.

Author reply: This article studied pre-activation to nucleate ice at lower relative humidity or higher temperatures compared to their intrinsic ice nucleation. This review presumes that ice preserved in pores is responsible for pre-activation and analyses pre-activation under this presumption. The authors stated that “*Therefore, pre-activation due to pore ice is constrained by the melting of ice in narrow pores and the sublimation of ice from wide pores imposing restrictions on the temperature and relative humidity range of pre-activation for cylindrical pores.*” Or “*This pre-activation mechanism is limited to high temperature by melting of ice in narrow pores.*” These statements show that when the liquid is in contact with walls which has activation sites and wall surface is concave, like the case that there is no oil, melting and freezing occur in higher temperatures in narrower pores which is due to the decreased ΔG^* . This is in harmony with our results (This case is similar to the scenario that we do not have any oil in the pore, **Fig. S10**).

* Marcolli, C. (2020). "Technical note: Fundamental aspects of ice nucleation via pore condensation and freezing including Laplace pressure and growth into macroscopic ice." *Atmos. Chem. Phys.* 20(5): 3209-3230.

Author reply: Also, in this article, the authors studied pore condensation and freezing (PCF) which is an ice nucleation mechanism. In PCF, liquid water condenses in pores of solid aerosol particles below water saturation, as described by the Kelvin equation, followed by homogeneous ice nucleation when temperatures are below 235 K or immersion freezing at higher temperatures, in case the pores contain active sites that induce ice nucleation. Also, the authors showed that the interfacial curvature through Laplace pressure affect the ice nucleation temperature. That is, as the liquid experiences negative pressure, the ice nucleation temperature increases. This is completely consistent with our findings shown in Fig. S11. We included this article in our references.

We discussed that if the droplet was on a rigid concave surface (i.e. water-wall interface), upon decreasing the size, ΔG^* decreases, due to the effect of surface factor on ΔG^* which is dominant compared to the effect of $\Delta\mu$ (See Figure S11). However, for soft interface of oil (i.e. concave surface of water-oil), this is not the case and the effect of surface factor becomes insignificant and what is important here is the effect of $\Delta\mu$. In this case, upon decreasing the size, ΔG^* increases as shown in **Figure 4b**. See equations below:

$$\Delta G^* = \frac{16\pi\gamma_{IW}^3}{3(\rho\Delta\mu)^2} f(m, x) \quad (1)$$

$$f(m, x) = \frac{1}{2} \left\{ 1 - \left(\frac{1+mx}{g_c} \right)^3 - x^3 \left[2 - 3 \left(\frac{x+m}{g_c} \right) + \left(\frac{x+m}{g_c} \right)^3 \right] + 3mx^2 \left(\frac{x+m}{g_c} - 1 \right) \right\} \quad (2)$$

$$g_c = (1 + x^2 + 2mx)^{1/2} \quad (3)$$

19) “Line 15: Oil should coat the walls, not wet the walls.”

Author reply: Wetting is the ability of a liquid to maintain contact with a solid surface, resulting from intermolecular interactions when the two are brought together. Wetting is a terminology in interfacial science community and coating is mostly used for hard solid coating (i.e. anti-corrosion coatings, anti-fouling coatings).

20) “Line 23-32: This argument presented here is interesting. But I think the authors are mixing up three different issues here and each should be addressed more clearly. One is the fact that the authors state that significant changes in the results are only observed if majority of pores are frozen. This strikes me as a limit of detection issue with their methodology and nothing to do with classical nucleation theory, or the theory of ice forming mechanisms. This should be separated from the next two points.

Next comes the aspects of the freezing at higher temperatures in the absence of an oil surrounding environment for the water. This means that the water is in direct contact with the pore walls and the authors say that the pores contain active sites, which I agree is completely fathomable. The active sites can catalyze ice germ formation, reduce the ΔG required for ice nucleation and thus increase the freezing temperature. Nonetheless it strikes me as interesting that in the case where the total water content in the membrane is not isolated, i.e. the water volume can be considered as a whole, since the water droplets are not isolated by oil layers in between their volume is also now not as restricted as in the oil case. So how can the authors say it is only due to the active sites, and ignore again the volume effect here?

Also, the freezing T being so high in the 2 – 4 nm pores suggests that the volume plays a role and that the volume of water to be taken into account is larger than that associated with the 2-4 nm dimension because at 273 K (the freezing temperature reported in the absence of oil) the size of an ice germ will be quite large. This calculation of the ice germ size as a function of freezing temperature must be presented in this work, in order to convince the reader that indeed water could freeze in the 2-4 nm confinements. The authors should also clarify that if there is no oil in the membrane, then is all the water volume considered continuous. i.e. is the water in one capillary in contact with another capillary. Based on how I understood the methods, the answer

to this question is yes, and the oil experiments are the only ones that ensure the water is isolated into smaller volumes.

So, while I agree, that active sites play a role, it cannot be the only reason based on the information presented here in.”

Author reply: On the first point, a large number of pores with the exact same dimension allows us to increase the sample pool size in our experiments and **reduce the error bar**. Statistically, these results are much more credible than probing an individual pore. This is one reason that we choose AAO membrane with uniform pore dimensions. It is not required to detect ice formation within each individual pore to show the effect of dimension on the freezing characteristics. We agree that the FTIR spectroscopy (beam spot) and resistivity method do not have accuracy to probe an individual pore.

On the second point, we mentioned that the active sites are *not* the only effect here and in addition to the effect of active sites, size (i.e. interfacial curvature) also matters. We discussed the effect of liquid volume above. The total volume of water cannot be considered as a whole, because although we do not have oil, the water droplets are isolated by pore walls. To achieve this, after filling the membrane with water, the membrane was dried with N₂ gas and was placed in environment for a while to evaporate surface water and only have the water inside the pores. The water inside the pores is a stable water due to the effects of curvature, nano-roughness, and confinement resulting in a lower vapor pressure for water and hence inhibiting its vaporization [Y. Tomo *et al.*, *Superstable ultrathin water film confined in a hydrophilized carbon nanotube*, *Nano Lett.* 2018, 18, 1869–1874]. The existence of water inside the pores is confirmed with FTIR. Our results (Figure S11) show the size dependence of nucleation temperature which means that there was no surface water that interconnects the water inside the pores. In addition, after filling with water and drying the surface water, QCM analysis performed on the membrane and the results (Table S2) show that the weight of water is close to the theoretical weight of water inside the pores that indicates there is no water on the surface of membrane that causes the water inside the pores to be interconnected.

On the third point, we should add the droplets are elongated ellipsoidal droplets with one diameter in the range of few nms and the other diameter in tens of micron. The high nucleation temperature of 2-4 nm in the absence of oil is due to the active sites and the curvature of water-air interface which induces a high negative pressure inside the water and increases chemical potential difference between the ice and water that leads to the reduction in ΔG^* [C. Marcolli, *Ice nucleation triggered by negative pressure*, *Sci. Rep.*, 7, 16634]. The calculation of the ice germ size as a function of freezing temperature is already presented in Table S3.

As explained above, also, in the absence of oil, the existence of isolated drops and discontinuity of water inside the membranes are proved.

Action: The first point is separated from the two other points.

22) “Page 7, Line 4: should read “..difference at the oil-water..”

Page 8

Line 1: “We did not observe..”

Line 3: should be nanodroplets

Line 5: should be clusters and replace “happens” with “occurs”

Line 6: “..temperatures in the range...”.”

Author reply: We changed the abovementioned sentences.

Action: Changes are made in the main manuscript.

23) *“Page 8, Line 11: The example given here by the authors of one developing spherical 2nm droplets in air would not be possible because the Kelvin effect would be so high, that the droplets would have no practical chance of surviving. They would require unrealistically high super saturations with respect to liquid water in order to survive. As such the example that follows about dropping the ice nucleation temperature to 213 K is hypothetical at best. Unless the authors clarify in which application can spherical 2 nm droplets actually be formed in a stable manner to use in any application.”*

Author reply: We agree that this may not possible, given instability of the droplet. We wrote as a hypothetical case.

Action: The above change is applied.

24) *“Figure 1: would be nice to have a legend indicating what is the oil layer here.*

Author reply: Oil layer is the layer around the water droplet and it is clarified on Figure 1 by a legend.

Action: The change is made on the figure in the main manuscript.

Figure 2: I-V should be defined as current and voltage. Since the main indicator of phase change is resistance here, why not plot V on the y-axis and I on the x-axis since the slope of this curve then gives the resistance and then it is easy to assess, higher slope means higher resistance and when there is a jump in the slope that is where the phase change occurs. That would be more intuitive and physical for the plots shown in Figure 2.

Author reply: In 4-point probing method, we apply current and read the voltage. Thus, we can adjust different currents and read the related voltage drop. In fact, 4-point probing method consists of four electrical probes in a line, with equal spacing between each of the probes. Based on this method, A high impedance current source is used to supply current through the outer two probes; a voltmeter measures the voltage across the inner two probes (See figures below). Due to the high impedance of voltmeter, no current flows through the inner electrodes. In this case the voltage drop is measured between two inner electrodes where wire resistances and contact resistances do not contribute in voltage measurement. Thus, we can adjust the current not the voltage and that is why we are using I-V curves. In addition, these I-V curves are in harmony with literature to show changes in **resistance or conductance** for each constant current [*For examples, Q. Xie et al. Figure*

2 in “Ion transport in graphene nanofluidic channels”, *Nanoscale*, 8, 47, 2016 or M. Nazari et al., *Surface Tension Nanogates for Controlled Ion Transport*, *ACS Appl. Nano Mater.* 2020, 3, 7, 6979–6986].

Figure 3: For the smaller pore dimensions it seems that the FTIR metrology gives slightly lower temperatures for the phase transition. Have the authors considered what local heating effects the FTIR beam would have on the pore volume water. i.e. when the water droplets are so small and isolated, the FTIR beam could be providing some local heating therefore requiring further cooling in order to freeze. Could this be a potential reason for the slightly lower T with the FTIR metrology for the smaller pore size water freezing?

Author reply: FTIR has a light source of 20 mW including a reference red light and infrared light ranging from 4000 cm^{-1} to 400 cm^{-1} . Based on the numerous experiments conducted in this FTIR platform and measured temperatures, these wavelengths can elevate the temperature of the sample for a negligible amount of less than $0.1\text{ }^{\circ}\text{C}$. As long as the system, including a ZnSe crystal stage, totally reflect the light and the maximum absorption is under 30% in our working range (Transmission curve is shown below), we expect that the temperature increase is minimal and within the error bar of our measurements.

Figure 9. Total transmission of 5mm thick ZnSe substrate

Supplementary Material

25) “Figure S1. Can the authors harmonise the style on the figures? In some of the figures the labels of the legend have different colors and sizes and the labels on the figures as well. Also, it is not clear what the difference between “height” and “height sensor” is here. What does height refer to in these figures?”

Author reply: The reason for the difference in the style on the figures is that these AFM images are captured through two different instruments and these are the images that are given to us by the software of the instrument.

26) “Figure S2: The authors often mention in the manuscript that the inner layer is the pore are concave, can this be schematically shown in this Figure. Also, can the dimension of 50 – 60 microns mentioned in the article also be depicted in this figure for clarity?”

Author reply: The concave surface of pore walls can be observed in **Figure 1b**. Regarding the concave interface of oil-water, due to the fact that ice nucleates inside the water not oil, the oil acts as a substrate on which the ice forms, as shown in figure below.

Figure 7. Schematic of water nano-droplet encapsulated in oil and the formation of ice embryo inside water.

Action: The thickness of membrane, which is 50-60 microns, is clarified in Figure S2.

27) “Figure S10: how are these results justified by CNT, without considering the total volume? It is the total active site density that matters, but that only matters if the active sites are all covered by a large volume of water. If the volume of water is really small, the probability it will cover an active site is also small. In this regard, the authors should demonstrate how an ice germ can form in a 2 nm pore at 273 K as demonstrated in this figure. In the caption the authors talk about curvature, I would expect because of water adhesive forces that the curvature is concave here. Can this be clarified as well?”

Author reply: We need to clarify that the water droplets inside the pores are elongated droplets with ellipsoid shape in which the smaller diameter is in the range of few nm while the other diameter is tens of micrometer. We mentioned that the active sites are not the only effect here and in addition to the effect of active sites, pore dimension (i.e. interfacial curvature induced by the smaller diameter) and Laplace pressure also matter, Fig. S11.

Also, here the total volume of water cannot be considered as a whole, because although we do not have oil, the water droplets are isolated by pore walls. To achieve this, after filling the membrane with water, the membrane was placed in the environment for a while to evaporate surface water and only have the water inside the pores. The water inside the pores is stable water due to the effects of curvature, nano-roughness, and confinement resulting in a lower vapor pressure for water and hence

inhibiting its vaporization [Y. Tomo *et al.*, *Superstable ultrathin water film confined in a hydrophilized carbon nanotube*, *Nano Lett.* 2018, 18, 1869–1874]. The existence of water inside the pores is confirmed with FTIR. In addition, our results (Figure S11) show the size dependence of nucleation temperature which means that there was no surface water that interconnects the water inside the pores. In addition, after filling with water and drying the surface water, QCM analysis performed on the membrane and the results (Table S2) show that the weight of water is close to the theoretical weight of water inside the pores that indicates there is no water on the surface of membrane that causes the water inside the pores to be interconnected.

The high nucleation temperature of 2-4 nm in the absence of oil is due to (1) active sites, (2) concave interface of water-wall, $f(m,x)$ and (3) the curvature of water-air interface which induces a high negative pressure inside the water and increases chemical potential difference between the ice and water that leads to the reduction in ΔG^* [C. Marcolli, *Ice nucleation triggered by negative pressure*, *Sci. Rep.*, 7, 16634]. The calculation of the ice germ size as a function of freezing temperature is already presented in Table S3.

Regarding the ice nucleation in 2 nm pores, we need to clarify that the water droplets inside the pores are elongated droplets with ellipsoid shape in which the smaller diameter is in the range of few nm while the other diameter is tens of micrometer.

Regarding the curvature, the curvature of water-wall interface where ice nucleates is concave as the reviewer mentioned when we have air around the confined water droplet (see figure below).

Figure 10. Water confined in the pores when there is no oil around that.

We again thank the reviewer for the thoughtful comments.

Reviewer #3:

“I agree that understanding the water/ice transformation is important in many different applications and that very few experiments can probe this transformation down to the nanoscale. Those that do often involve aerosols or large molecular clusters that are rapidly cooled in supersonic or free-jet expansions, although recent work has also focused freezing in nanopores and nano-cages – both hydrophilic and hydrophobic. The use of the membranes to confine the lengthscale of the “droplets” is most similar to that of freezing in hydrophobic pores. Unfortunately, the current data are not compared to any of the recent literature that have studied freezing under confinement that use different measurement techniques. Nor does it try to relate the results to extensive literature regarding emulsion droplet freezing (see Refs 8,9, 10 and refs therein). Thus, the uniqueness and importance of the current work is not established, and the key point of the paper is not clear. Some of the claims and assumptions made in the paper are neither demonstrated nor referenced.”

We elaborated the context of this work and connection with the other works in the comments below. Just as a synopsis, the novelty of this work is that (1) for just water-filled pores, as one moves to few nm scales concave surfaces, the ice nucleation temperature increases (0°C for 2 nm water droplet). However, once oil encapsulates these droplets, the trend completely changes, and ice nucleation temperature decreases (-41°C for 2 nm water droplet encapsulated with oil). Note that for both of these droplets, the ice nucleating surface is concave. Also, the Laplace pressure effect by itself (i.e. through its role on chemical potential) cannot justify this contrast in the findings. Furthermore, (2) contrary to nucleation on curved stiff solid surfaces, ice formation on soft interfaces (omnipresent in nature) could deform the interface leading to suppression of ice nucleation. We have provided even further experiments to support point (2).

Comments:

1) *“The authors claim that the ice that forms is hexagonal even down to 2 nm without the definitive proof provided by a diffraction experiment. This result conflicts directly with the significant experimental and theoretical literature that finds that ice formed by freezing from supercooled water (homogeneously or heterogeneously) forms stacking disordered ice, ice Isd, that only anneals to hexagonal ice on the time scale of hours [11-13]. Although recent Raman experiments [14] have been able to distinguish Isd from Ih, to this reviewer’s knowledge the same is not true for IR experiments. The work of Bertie and Whalley [15] states that hexagonal and cubic ice have IR spectra that are indistinguishable within the measurement limits. The reviewer is not aware of any papers that correct this view, and the authors do not provide evidence or even a reference. They simply state (without reference) that “The formed ice phase in the few nm droplets shows OH stretch peak of 3200 cm⁻¹ rather than 3080 cm⁻¹ (cubic) suggesting that all of the ice phases formed in the nanodroplets are Ih”. This statement is also contradicted by the work of Amaya et al. [16] and Amaya & Wyslouzil [3] that showed droplets ~20 nm in diameter that froze near 225 K were highly cubic stacking disordered ice I (based on diffraction measurements) but their FTIR measurements look very similar to those shown here.”*

Author reply: Thanks for this helpful comment. The interpretation of our data was based on the Laplace pressure in the droplets and FTIR peak with the following reference (Figure S3).

[D. Shin et al., *Ice-VII-like molecular structure of ambient water nanomeniscus*, *Nat. comm.* (2019)]

Based on reviewer's comment and the aforementioned reference, there was a misinterpretation of our data and we may have stacking disordered ice with a high degree of cubicity. Also, it should be noted that the other parameter which can affect the structure of ice is the geometry and dimension of confinement. In fact, ice forms in a structure in a way that its lattice can fit the confinement [Y. Bi et al., *Enhanced heterogeneous ice nucleation by special surface geometry*, *Nat. Comm.* (2017)]. Considering these arguments, interpretation on exact type of ice needs further experiments and analysis. As the purpose of this paper is about the effect of confinement and soft interfaces on ice nucleation, we keep this question open for future studies.

Action: We included a sentence in the manuscript to highlight possible high degree of cubicity in the formed ice phase and kinetically controlled transition to Hexagonal ice.

[11] Murray, B. J.; Knopf, D. A.; Bertram, A. K. *The formation of cubic ice under conditions relevant to Earth's atmosphere*. *Nature* 2005, 434, 202-205:

They present laboratory experiments demonstrating that cubic ice forms along with hexagonal ice when micrometer-sized droplets of pure water and aqueous solutions freeze homogeneously at cooling rates approaching those found in the atmosphere. They found that the formation of cubic ice is dominant when droplets freeze at temperatures below 190 K.

[12] Benjamin J. Murray and Allan K. Bertram*, *Formation and stability of cubic ice in water droplets*, *Phys. Chem. Chem. Phys.*, 2006, 8, 186–192:

They studied the formation of cubic ice in pure water droplets suspended in an oil matrix as a function of droplet size. The results show that droplets of volume median diameter 5.6 μm froze dominantly to cubic ice with stacking faults. Their results indicate that cubic ice is the crystalline phase that nucleates when pure water droplets freeze homogeneously. They also showed that as the size of the water droplets increased, the formation of the stable phase of ice, hexagonal ice, was favored.

[13] Tamsin L. Malkin, Benjamin J. Murray, Christoph G. Salzmann, Valeria Molinero, Steven J. Pickering and Thomas F. Whale, *Stacking disorder in ice I*, *Phys.Chem.Chem.Phys.*, 2015, 17, 60:

They showed that ice which crystallizes after heterogeneous nucleation in water droplets containing solid inclusions also contains stacking disorder even at freezing temperatures of around -15°C . The structure of ice that crystallizes initially from supercooled water is always stacking-disordered and that this metastable ice can transform to the stable hexagonal phase subject to the kinetics of recrystallization.

[16] Amaya, A. J.; Pathak, H.; Modak, V. P.; Laksmono, H.; Loh, N. D.; Sellberg, J. A.; Sierra, R. G.; McQueen, T. A.; Hayes, M. J.; Williams, G. J.; Messerschmidt, M.; Boutet, S.; Bogan, M. J.; Nilsson, A.; Stan, C. A.; Wyslouzil, B. E. *How Cubic Can Ice Be?* *Journal of Physical Chemistry Letters* 2017, 8, 3216-3222:

They investigated the crystal structure of ice formed by homogeneous ice nucleation in deeply supercooled water nanodrops ($r \approx 10 \text{ nm}$) at $\sim 225\text{K}$. The X-ray diffraction spectra indicate that

this ice has a metastable, predominantly cubic structure; the shape of the first ice diffraction peak suggests stacking-disordered ice with a cubicity value, χ , in the range of 0.78 ± 0.05 .

[3] Amaya, A. J.; Wyslouzil, B. E. *Ice nucleation rates near 225 K. Journal of Chemical Physics* 2018, 148 Art No. 084501:

They measured the ice nucleation rates, J_{ice} , in supercooled nano-droplets with radii ranging from 6.6 nm to 10 nm and droplet temperatures, T_d , ranging from 225 K to 204 K. They stated that under these conditions the solid particles are stacking disordered ice with a high degree of cubicity. In addition, their FTIR peaks are similar to measured FTIR peaks in this work.

Action: We included some of these literatures and clarified confusion on the ice phase formed in these nanopores.

2) “The FTIR experiments are not very well described in the main text or in the SI. There is no figure illustrating the setup. How was the temperature controlled? How well is the temperature known? How was water vapor prevented from condensing from the atmosphere? Where are the C-H stretch bands from the oil? Were they suppressed or subtracted? What was the resolution used to make the measurements?”

Author reply: The FTIR experimental setups are shown below. We used peltier coolers to control the sample temperature. In Attenuated Total Reflectance (ATR) mode, once the sample was placed on the FTIR stage, the peltier coolers were placed on a part of the membrane covered by a coverslip. For probing the sample temperature, a thermocouple was attached on the edge of the membrane. Because the membrane is placed between coolers, FTIR stage and coverslip, the vapor condensation from the surrounding environment is minimal. Furthermore, in order to prevent any possible frost formation, nitrogen gas was purged in the setup continuously. In the transmission mode, the membrane was sandwiched between two coverslips and mounted on a stand and the light passed through the sample.

C-H peaks from oil are subtracted using background and the resolution of the instrument is 4 cm^{-1} .

Figure 11. Schematic of FTIR analysis setup that is used.

Action: Changes are made in FTIR part in the material and method as well as SI sections.

3) “The authors claim (without proof) that nucleation is heterogeneous and occurs at the oil water interface. To prove this requires experiments that demonstrate nucleation rates scale with available surface area or available volume. This paper does not characterize the water domains beyond stating that maximum radius is set by the tube diameter. After reading of the SI + the paper, it sounds more like the water is in tube-like structures (rather than spherical drops) with some constrained radius and variable/ unknown length. Without characterizing the domains, how can the current experiments prove what the nucleation mechanism is or whether it changes as a function of pore size?”

There is a great deal of literature that discusses homogenous nucleation in water droplets surrounded by oil/surfactant mixtures [8-10]. Most of these are not consistent with nucleation occurring at the oil-water interface, but rather with nucleation throughout the droplet. Why is this case different? How can that be proven?”

Author reply: We will discuss the referred literature first and after address the comments.

[8] Riechers, B.; Wittbracht, F.; Huetten, A.; Koop, T. The homogeneous ice nucleation rate of water droplets produced in a microfluidic device and the role of temperature uncertainty. *Physical Chemistry Chemical Physics* 2013, 15, 5873-5887:

The authors experimentally investigated ice nucleation in water droplets with diameters between 53 and 96 micrometers. It is stated that homogeneous ice nucleation becomes efficient (typically 235–238 K depending on the cooling rate and droplet size) and this statement is based on the studied temperature here.

[9] Atkinson, J. D.; Murray, B. J.; O’Sullivan, D. Rate of Homogenous Nucleation of Ice in Supercooled Water. *Journal of Physical Chemistry A* 2016, 120, 6513-6520:

The authors examined the freezing of pure water droplets with diameters between 4 and 24 μm . Under the assumption that nucleation occurs within the bulk of the droplet, they concluded that at

smaller sizes ($<6 \mu\text{m}$) they cannot rule out a significant contribution of surface nucleation which is in harmony with our findings.

[10] Mark D. Tarn, Sebastien N. F. Sikora, Grace C. E. Porter, Daniel O'Sullivan, Mike Adams, Thomas F. Whale, Alexander D. Harrison, Jesús Vergara-Temprado, Theodore W. Wilson, Jung-uk Shim, Benjamin J. Murray, *The study of atmospheric ice-nucleating particles via microfluidically generated droplets, Microfluidics and Nanofluidics (2018) 22:52:*

The authors presented the application of microfluidic devices to the study of atmospheric INPs via the simple and rapid production of monodisperse droplets and their subsequent freezing on a cold stage. They stated that in the absence of nucleation sites, micron-sized water droplets freeze homogeneously at around $-38 \text{ }^\circ\text{C}$ on typical laboratory timescales.

All of the above references distinguished heterogeneous and homogeneous nucleation based on the working temperature. That is, if the temperature is around 235 K, the nucleation occurs homogeneously. Here, we have distinguished these modes of nucleation through Gibbs energy barrier (ΔG^*) analysis. If the nucleation in these pores were homogenous, we should not observe any effect of confinement size on the nucleation temperature. For example, for the case of pure water in the confinement, the nucleation temperature varies from -7 to 0°C , Fig. S10.

To further confirm this, we performed another set of experiments with membrane with pore dimension of 10 nm where we switched the encapsulating Octane phase with PDMS with various modulus as shown in **Table S6**. In these experiments, we initially filled the membranes with water. Mixtures of Sylgard (i.e. ratio of base to crosslinker) and octane (as solvent) with three concentrations are developed. Each mixture is applied to the water filled membranes and allowed to be cured for 24 hrs at room temperature. Note that Octane evaporates during this period. That is, the water droplet are encapsulated with PDMS with different modulus. After, we performed FTIR analysis in transmission mode and the results are shown in **Fig. S15**. The results show that as the modulus of interface increases, due to the effect of surface factor on Gibbs energy barrier, ice nucleation temperature increases. This is another proof that the ice nucleation is heterogenous and not within the bulk of the liquid.

This additional experiment also confirms that the soft octane-water interface leads to suppression of freezing to extremely low temperatures.

Figure 12. TIR spectrum of nanodroplets in membrane with pore diameter of 10 nm surrounded by PDMS/Octane with different moduli (different ratio of prepolymer to curing agent) (a) 10:1 (b) 30:1 and (c) 60:1

Table 3. Nucleation temperature of water nano-droplet confined in 10 nm nanopores and encapsulated with PDMS/Octane with different moduli.

Base: Cross linker (weight ratio)	Young modulus (kPa)	T _N (°C)
10: 1	1300	-9
30: 1	120	-9
60: 1	3	-15
Just Octane	0	-26

[Michael L Smith, Delphine Gourdon, William C Little, Kristopher E Kubow, R. Andresen Eguiluz, Sheila Luna-Morris, Viola Vogel, Force-Induced Unfolding of Fibronectin in the Extracellular Matrix of Living Cells, PLoS BIOLOGY, Vol. 5 (2007).]

Action: The new experiments are added to the manuscript.

4) “The statement that the temperature for freezing 2 nm droplets drops below the “bulk homogeneous nucleation limit” is correct, but it is not surprising. Aerosol droplets in the same size range do not nucleate until the temperature is well below this limit as well.1-3 It is not clear what is special here. Given that the claim is that nucleation at the interface will reduce the energy barrier to nucleation, shouldn’t the comparison be to other droplets or water domains under comparable conditions and sizes rather than to bulk (micron sized)?”

Author reply: Here, we compare water droplets in pores with and without encapsulating oil. The nucleation temperature of pure water in the nanopores is shown in **Fig. S10**. As shown, in these droplets the nucleation temperature increases as one moves to few nm pore dimensions (-7°C for 20 nm pores and 0°C for 2 nm pore). However, as oil phase is introduced here, the trend unexpectedly changes and for few nm droplets in the pore, the nucleation temperature decreases. (-42°C for 2 nm pore). This is entirely due to the soft oil-water interface. Furthermore, we added another section in which we showed that by tuning modulus of the oil, the nucleation temperature increases.

Action: The new section is added to the manuscript.

5) “The authors chose octane to maximize the interfacial tension and to thereby ensure a high internal droplet pressure. They then add surfactant to the oil to improve electrical conductivity without discussing any possible effects this may have on interfacial tensions and, thus pressure. Nor do they discuss the effect that pressure can have on the predictions of classical theory although this has been discussed extensively in the literature especially by Espinoza and co-workers.”

Author reply: On the second point, we have extensively discussed the role of Laplace pressure on CNT predictions: Eq 2, 3 ($p\Delta v$ term) and the role of $p\Delta v$ on $\Delta\mu_{Nd}$ as shown in Fig. 4(a). All the pressures are listed in Table S1.

In fact, all of our data interpretation is based on considering the effect of Laplace pressure. As shown in the plot below, as size decreases Laplace pressure can increase significantly. In this case, the parameter $P\Delta v$ becomes significant where it changes $\Delta\mu$ (Eq. 2 in the manuscript) and as a

result Gibbs energy barrier for nucleation (Eq. 1 in the manuscript). The effect of pressure is considered in all of those calculations.

Figure 13. Positive pressure induced inside the water nano-droplets encapsulated by oil due to the interfacial curvature and Laplace pressure.

On the first point, HLB (Hydrophilic–lipophilic balance) value for span 80 is 4.3 which means that span 80 is oil soluble and water insoluble (*Y. Lo, Relationships between the hydrophilic–lipophilic balance values of pharmaceutical excipients and their multidrug resistance modulating effect in Caco-2 cells and rat intestines, Journal of Controlled Release 90, 37-48 (2003)*). Furthermore, we used the pendant drop method to access the role of Span80 concentration on octane-water interfacial tension. In these experiments, a water droplet is introduced in an octane medium with different concentrations of Span80 as shown below. The Bond number (Bo) is defined as

$$Bo = \frac{\Delta\rho g R_0^2}{\gamma_{ow}}$$

Where $\Delta\rho$ denotes the density difference between water and octane, g is the gravitational acceleration and R_0 is the radius of curvature at the drop apex, **Figure 3**. The differential form of Young-Laplace equation in terms of arc length (s) is written as

$$\begin{aligned} \frac{d\phi}{d\bar{s}} &= 2 - Bo \bar{z} - \frac{\sin \phi}{\bar{r}} \\ \frac{d\bar{r}}{d\bar{s}} &= \cos \phi \\ \frac{d\bar{z}}{d\bar{s}} &= \sin \phi \end{aligned}$$

where the bar indicates dimensionless values scaled by R_0 . The boundary conditions are

$$@ s = 0: \bar{r} = 0; \bar{z} = 0; \phi = 0$$

The input to this system of equations is Bo or specifically γ_{ow} .

Figure 3. Schematic of pendant drop experiments.

Once this value is given, one can solve this system of equations numerically. We determined numerically the shape of the droplet for a range of surface tension values and compared the calculated droplet shape with the one measured, **Figure 4**. For each considered value of surface tension, we determined the coefficient of determination (R^2) between the calculated shape and measured shape. The surface tension with the highest value of R^2 provides gives us γ_{ow} through pendant droplet method, **Table 1**. As a control experiment, we compared the measured interfacial tension for octane-water with the reported value in the literature which shows less than 5% error.

Figure 4. The measured shape of water droplet suspended in octane are compared with the calculated shape to determine the surface tension of octane-water interface as a function of Span80 concentration. Note that even up to the concentration of 4500 ppm, the changes in surface tension is less than 5%.

Table 1: Surface tension of oil-water interface as a function of Span80 concentration

Concentration of Span 80	γ_{ow} in literature (mNm^{-1})	Measured γ_{ow} (mNm^{-1})
0 ppm	51	50
10 ppm		49
50 ppm		51
250 ppm		50
4500 ppm		48

[17] Espinosa, J. R.; Zaragoza, A.; Rosales-Pelaez, P.; Navarro, C.; Valeriani, C.; Vega, C.; Sanz, E. *Interfacial Free Energy as the Key to the Pressure-Induced Deceleration of Ice Nucleation. Physical Review Letters* 2016, 117 Art No.135702:

The authors performed simulations to investigate ice nucleation at high pressures consisting in embedding ice seeds in supercooled water. They found that the slowing down of the nucleation rate is mainly due to an increase of the ice I-water interfacial free energy with pressure. They stated that

applying pressure significantly increases the range of temperatures at which liquid water may exist. Also, they stated that the increase of the ice I-water interfacial free energy is the main reason for the decelerating effect of pressure on ice nucleation (at least up to 2000 bar).

Their work provides a physical explanation to the high pressure freezing techniques used in the preservation of food and biological samples. In fact, we induced high pressure by inducing large curvature at water-oil interface.

[18] Jorge R. Espinosa, Carlos Vega, and Eduardo Sanz, Homogeneous Ice Nucleation Rate in Water Droplets, Phys. Chem. C 2018, 122, 22892–22896:

The authors stated that the pressure inside nanoscopic drops is larger than the atmospheric one by virtue of the Laplace equation. In this work, they considered such pressure rise to predict the nucleation rate in droplets using the TIP4P/Ice water model. They evaluated the pressure inside the drops with the Laplace equation. Then, they obtained the nucleation rate as a function of the supercooling by interpolating their previous results for 1 and 2000 bar using the classical nucleation theory. This is complete harmony of our analysis in this manuscript.

Action: We included these references in the manuscript.

6) *“The authors show data for water in contact with pores freezing at ~ -7 C for 20 nm pores, and water surrounded by oil in 150 nm pores freezing at ~ -10 C. What would be more inciteful would be experiments in the 150 nm pores in the absence of oil to tie the two types of experiments together.”*

Author reply: Thank you. We performed the additional experiments for other sizes (150, 80 and 40 nm) and nucleation temperature for all of them is around -7°C which is the same as 20 nm. This indicates that the effect of curvature becomes significant in dimensions below 10 nm.

Figure 14. FTIR spectrum of nanodroplets in various pore dimensions when pores are filled with water without addition of oil around them. (a) For 150 nm nanodroplets, at temperature of 7°C, the OH stretch peak is narrowed and red-shifted to wavenumber of ~3200 cm⁻¹. This red shift indicates the water-ice phase change and provides TN value at this length scale. (b) The red shift occurs at temperature of -7°C for 80 nm water droplets as well. (c) The red shift occurs at temperature of -7°C for 40 nm water droplets as well. (d) The nucleation temperature of water droplets, when the membrane is filled with water without addition of oil, measured through FTIR metrology is shown as a function of length scale.

Action: The curves are added to the supplementary information.

This reviewer is not familiar with the electrical resistance methods used by the authors, but the results appear to be consistent with the FTIR measurements. The reviewer is familiar with FTIR measurements used to follow freezing and with nucleation theory.

We again thank the reviewer for the thoughtful comments.

REVIEWER COMMENTS

Reviewer #1 (Remarks to the Author):

I thank the authors for their detailed response and appreciate their efforts made, both concerning the additional experiments as well as the edits to the manuscript. Most of my questions have been addressed. However, I still have the following concerns related to the electrical measurements and their interpretation:

1) The temperature range of the control experiment performed on octane-filled pores in Fig. S10 to me is not convincing. For water filled pores of the same size, the nonlinearity occurred between -12 C and -14 C (Fig. S9a). Why stop the control experiment at -14 C when even for water-filled pores of the same size the authors explored lower temperatures? The goal of a control experiment is to credibly rule out the presence of non-linearities in the resistance in the absence of water. The authors state that liquid-solid phase change of octane occurs only at -57 C, but they don't demonstrate that above this temperature octane itself does not cause non-linearities such as they attribute to water freezing. Fig. 2e suggests that temperatures down to -45 C are readily accessible for the authors. Doing a control experiment down to this temperature will significantly strengthen their case.

2) To clarify: when I asked whether I/V curves are symmetric around zero, I meant "symmetric about the origin". Are I/V curves symmetric about the origin and if not, do non-linearities occur at the same temperature when the polarity is reversed? I'd expect membranes as shown in Fig. S2b to yield I/V curves symmetric about the origin and only membranes of type S2a to possibly yield non-symmetric I/V curves, but non-linearities should occur at the same temperature. This is a consistency check.

3) The authors mention that as water freezes, its electrical resistance increases by three orders of magnitude, from kOhm to MOhm in their case. Can they elaborate on how they determine these resistances? This goes back to my original question about where the authors think the voltage drops occur. I understand the 4-point probe measurement. What I was trying to get at is an equivalent circuit for charge transport through a pore. A change from kOhm to MOhm does not seem to be enough to explain the large jumps on the GOhm scale observed at the non-linearities in their experiment. This would in fact be clearer if the y-axis in Fig. 2e had units of Ohm.

Reviewer #2 (Remarks to the Author):

The authors seemed to have addressed sufficiently most of my previous comments. Below are the issues I still consider outstanding in the current version of the manuscript, which should be addressed before publication can be recommended.

4) The changes made are insufficient, and my previous comment still stands. The authors write “homogeneous nucleation of nanoscopic liquid water droplets is ...” this still implies that the study explores nucleation of water droplets, but the study is investigating homogeneous freezing of nanoscopic water, thus the statement should read “homogeneous freezing of nanoscopic liquid water droplets is ...”

9) “(see Fig. 1a)”

11) This is indeed clearer but then I suggest authors refer to pore diameter, membrane diameter and pore length or membrane thickness (which represents pore length)

17) This illustration helps. the description of concave could be added to the SI where it is Figure S20. The concavity is with respect to viewing from inside the droplet. Typically, concavity and convexity is described with respect to viewing from above the surface of a droplet, in this regard the water in the oil would be convex. I view this as a simple definition problem with my reference point being above the surface of the droplet.

18) Marcolli 2017. I disagree with the authors view that this paper is in harmony with the results highlighted. If the The pore ice in narrow pores melts at lower temperatures than pore ice in larger pores. i.e. the freezing point depression is much higher in narrower pores than in larger pores. But the authors are suggesting the opposite. Indeed Marcolli 2017 shows that if particles were warmed up between successive ice nucleation cycles, the pre-activation effect was lost, and this is because of pore ice melting by the temperature being increased or sublimated due to a decrease in RH. I believe the authors should carefully read the article. The quote taken out of the conclusions should be presented in context of the main body of the paper. For example, “*The fraction of particles that remain pre-activated decreases with increasing temperature*” goes directly against what the authors have written in their rebuttal. Also, the notion of ice surviving or not melting up to temperatures close to 273 K is specifically for swelling pores which is not the case here in this study. As such I strongly recommend the authors to reconsider this evaluation. See also for example also from Marcolli 2017 “*Since pre-activation due to pore ice has bounds to small pore diameters given by the melting point depression in confinement and to large diameters due to vaporation at low RH, swelling pores seem to be best suited for persistent preactivation.*”

Reviewer #3 (Remarks to the Author):

Despite my comments in the previous round of reviews, adding references that clearly point out that this is not true, and claiming to have fixed these issues, the Manuscript still contains the following incorrect claims.

(1) regarding the phase of ice formed

near the bottom of page 2:

“The ice phase formed within nanodroplets as small as 2 nm in diameter is Ih (hexagonal).”

In their rebuttal letter they claimed to have fixed this misconception i.e. their letter states:

“Action: We included a sentence in the manuscript to highlight possible high degree of cubicity in the formed ice phase and kinetically controlled transition to Hexagonal ice.”

Although I did find the sentence consistent with this “action” leaving the sentence on page 2 is does not revise the manuscript adequately.

(2) the ability of FTIR to determine the phase of the ice.

the revised manuscript still claims

“We found that the OH stretch bond in these few nm nanodroplets is similar to the larger nanodroplets and indicates that the ice phase formed in these small nanodroplets is Ih phase.”

Since FTIR cannot distinguish between ice Ih, Ic or stacking default, this statement is incorrect. Only diffraction data would be able to tell if this were true or not.

Revision Report

Reviewer #1:

I thank the authors for their detailed response and appreciate their efforts made, both concerning the additional experiments as well as the edits to the manuscript. Most of my questions have been addressed. However, I still have the following concerns related to the electrical measurements and their interpretation:

We thank the reviewer for the insightful/constructive comments.

1) “The temperature range of the control experiment performed on octane-filled pores in Fig. S10 to me is not convincing. For water filled pores of the same size, the nonlinearity occurred between -12 C and -14 C (Fig. S9a). Why stop the control experiment at -14 C when even for water-filled pores of the same size the authors explored lower temperatures? The goal of a control experiment is to credibly rule out the presence of non-linearities in the resistance in the absence of water. The authors state that liquid-solid phase change of octane occurs only at -57 C, but they don't demonstrate that above this temperature octane itself does not cause non-linearities such as they attribute to water freezing. Fig. 2e suggests that temperatures down to -45 C are readily accessible for the authors. Doing a control experiment down to this temperature will significantly strengthen their case.”

Author reply: We thanks again the reviewer for this insightful comment. We conducted these experiments down to -32 °C and **Supplementary Figure 11** is updated. (Please see below) As shown, no nonlinearity is observed in the control experiments. The nonlinearity in case of water-ice transformation for this pore diameter was observed at -12 °C.

Supplementary Figure 11. I-V curves measured across the membrane filled with Octane when there is no water inside the pores with pore diameter of 80 nm. There is no nonlinear jump in electrical resistivity down to -32°C.

Action:

The above plot is added to the supplementary information.

2) “To clarify: when I asked whether I/V curves are symmetric around zero, I meant “symmetric about the origin”. Are I/V curves symmetric about the origin and if not, do non-linearities occur at the same temperature when the polarity is reversed? I’d expect membranes as shown in Fig. S2b to yield I/V curves symmetric about the origin and only membranes of type S2a to possibly yield non-symmetric I/V curves, but non-linearities should occur at the same temperature. This is a consistency check.”

Author reply: We conducted the I-V experiments for pore dimension of 80 nm in two cases: (1) pure Octane case at two temperatures of -10 and 25 °C and (2) for water droplets surrounded by the Octane. The results for case (1) are shown in **Supplementary Figure 12** suggesting Ohmic behavior. The results for case (2) are shown in **Supplementary Figure 13** that indicate the nonlinearity in the system resistivity is observed in both positive and negative domains with Ohmic characteristics. Note that the minimum current value with the source meter system is 10 pA and we have high uncertainty in low current region.

Supplementary Figure 12: The complete I/V curve for a membrane with pore diameter of 80 nm filled only with Octane at two temperatures which indicates the ohmic characteristic of this system.

Supplementary Figure 13: The complete I/V curve for a membrane with pore diameter of 80 nm filled with water surrounded by Octane. The temperature of the system is gradually decreased to probe water-ice transformation temperature. The nonlinearity in the resistance indicates phase transformation temperature. The system shows an ohmic characteristics.

Action:

The above plots and descriptions are added to the supplementary information.

3) *“The authors mention that as water freezes, its electrical resistance increases by three orders of magnitude, from $k\Omega$ to $M\Omega$ in their case. Can they elaborate on how they determine these resistances? This goes back to my original question about where the authors think the voltage drops occur. I understand the 4-point probe measurement. What I was trying to get at is an equivalent circuit for charge transport through a pore. A change from $k\Omega$ to $M\Omega$ does not seem to be enough to explain the large jumps on the $G\Omega$ scale observed at the non-linearities in their experiment. This would in fact be clearer if the y-axis in Fig. 2e had units of Ω .”*

Author reply: This comment helped us to clarify a point on electrical resistivity of water and ice. The value that we reported in $k\Omega$ and $M\Omega$ are total resistance of water and ice that we formed in a Petri dish. The Petri dish was filled with water solution and the electrode were placed in the water to measure electrical resistance. After transforming water to ice, we repeated the electrical resistivity measurements.

To be accurate, we needed to specify the specific resistivity of water and ice. We took two independent approaches to show that the electrical resistivity of water and ice could differ by more than 3 orders of magnitude. Also, the y-axis is intentionally designed unitless to be able to include all the graphs in one curve. The important feature that we are looking in this graph is the temperature at which nonlinearity occurs.

Approach 1: As mobility of ions in ice approaches zero, the resistance of ice is close to its pure ice value as reported as 10^7 ohm.m. (C. Jaccard, Mechanism of the electrical conductivity in ice, Annals of the New York Academy of Sciences, 125, 390-400, 1965). The electrical resistivity of water with 50 ppm of salt is reported ~ 100 ohm.m (Steve Felber, Water Fundamentals Handbook, DRI-STEEM, 2017). This suggests that there is five orders of magnitude difference between specific resistivity of water solution and pure ice. Please note that the membrane is made of many pores surrounded by water/octane and calculation of total resistance of the system will have high uncertainty.

Approach 2: In the second approach, we measured the electrical resistance of water solution and ice filled in a plastic tube with inner diameter of 1 mm as shown below. The tube was filled with water solution with 50 ppm NaCl and four-probe electrodes were attached to the tube. The specific electrical resistivity of the water is measured at 1°C as 395 ohm.m. The temperature of the system is dropped to -10°C allowing to ice form in the tube. The specific electrical resistivity of ice was measured as 2093333 ohm.m. That is, the specific resistivity of ice is ~ 5000 times higher than that of water solution.

Figure S11. The specific electrical resistivity of water solution and corresponding ice phase were measured in a plastic tube with inner diameter of 1 mm.

Action:

The above plot and approaches are included in the supplementary information.

Reviewer #2:

“The authors seemed to have addressed sufficiently most of my previous comments. Below are the issues I still consider outstanding in the current version of the manuscript, which should be addressed before publication can be recommended.”

We thank the reviewer for the insightful/constructive comments.

1) *“The changes made are insufficient, and my previous comment still stands. The authors write “homogeneous nucleation of nanoscopic liquid water droplets is ...” this still implies that the study explores nucleation of water droplets, but the study is investigating homogeneous freezing of nanoscopic water, thus the statement should read “homogeneous freezing of nanoscopic liquid water droplets is ...”*

Action: The modification is applied.

2) *“(see Fig. 1a)”*

Action: The modification is applied.

3) *“This is indeed clearer but then I suggest authors refer to pore diameter, membrane diameter and pore length or membrane thickness (which represents pore length)”*

Action: The modification is applied.

4) *“This illustration helps. the description of concave could be added to the SI where it is Figure S20. The concavity is with respect to viewing from inside the droplet. Typically, concavity and convexity is described with respect to viewing from above the surface of a droplet, in this regard the water in the oil would be convex. I view this as a simple definition problem with my reference point being above the surface of the droplet.”*

Action: We included the definition of concavity in the SI.

5) *“Marcolli 2017. I disagree with the authors view that this paper is in harmony with the results highlighted. If the The pore ice in narrow pores melts at lower temperatures than pore ice in larger pores. i.e. the freezing point depression is much higher in narrower pores than in larger pores. But the authors are suggesting the opposite. Indeed Marcolli 2017 shows that if particles were warmed up between successive ice nucleation cycles, the pre- activation effect was lost, and his is because of pore ice melting by the temperature being increased or sublimated due to a decrease in RH. I believe the authors should carefully read the article. The quote taken out of the conclusions should be presented in context of the main body of the paper. For example, “The fraction of particles that remain pre-activated decreases with increasing temperature” goes directly against what the authors have written in their rebuttal. Also, the notion of ice surviving or not melting up to temperatueres close to 273 K is specifically for swelling pores which is not the case here in this*

study. As such I strongly recommend the authors to reconsider this evaluation. See also for example also from Marcolli 2017 “Since pre-activation due to pore ice has bounds to small pore diameters given by the melting point depression in confinement and to large diameters due to vaporation at low RH, swelling pores seem to be best suited for persistent preactivation.”

Author reply: Thank you for the clarification. We studied the article in detail. The article discusses the conditions for pre-activation of pores for ice nucleation and suggests that the freezing point depression is higher for smaller pores.

In this manuscript, we did not induce any pre-activation state in the pores and observed *the opposite* trend of freezing point depression. The main physic of freezing point depression here is concavity of the pore. The pre-activation of pores by preserved ice is definitely another route to tune freezing point depression.

Reviewer #3:

“Despite my comments in the previous round of reviews, adding references that clearly point out that this is not true, and claiming to have fixed these issues, the Manuscript still contains the following incorrect claims.

We thank the reviewer for the insightful/constructive comments.

(1) regarding the phase of ice formed near the bottom of page 2:

“The ice phase formed within nanodroplets as small as 2 nm in diameter is Ih (hexagonal).”

In their rebuttal letter they claimed to have fixed this misconception i.e. their letter states:

“Action: We included a sentence in the manuscript to highlight possible high degree of cubicity in the formed ice phase and kinetically controlled transition to Hexagonal ice.”

Although I did find the sentence consistent with this “action” leaving the sentence on page 2 is does not revise the manuscript adequately.

Action: We apologize for this overlook. We corrected the mistake in the manuscript. It read now “The ice phase formed within nanodroplets as small as 2 nm in diameter is possibly stacking default ice (Isd) and could transition to Hexagonal ice (Ih) in a slow kinetically-controlled manner.”

3) *“the ability of FTIR to determine the phase of the ice. the revised manuscript still claims*

“We found that the OH stretch bond in these few nm nanodroplets is similar to the larger nanodroplets and indicates that the ice phase formed in these small nanodroplets is Ih phase.”

Since FTIR cannot distinguish between ice Ih, Ic or stacking default, this statement is incorrect. Only diffraction data would be able to tell if this were true or not.

Action: We apologize for this overlook. We corrected the mistake in the manuscript. It read now “The OH stretch bond in these few nm nanodroplets is similar to the larger nanodroplets, but FTIR metrology is not capable of distinguishing between Ih, Ic and Stacking default ice (Isd).”

REVIEWER COMMENTS

Reviewer #1 (Remarks to the Author):

The authors have addressed all of my concerns and the paper is acceptable for publication from my perspective.

Reviewer #2 (Remarks to the Author):

Dear Authors,

I still have one outstanding issue to be addressed in the manuscript. Please see attachment. I think this one point needs to be acknowledged in the manuscript and explained, in particular because it also is a novelty of the study.

Reviewer 2

Reviewer #3 (Remarks to the Author):

no more comments

Revision Report

Reviewer #2:

"I now agree with the authors, that indeed the authors observed the opposite trend of freezing point depression. Which was my point in the original review, that this was not sufficiently acknowledged and not explained.

The authors acknowledge this in the review process, but they really need to explain this opposite trend in the paper and explain why they observe this opposite trend. The published literature shows that with smaller pore sizes the freezing point of the pore ice is depressed, if they see an opposite trend, this needs to be explained in my opinion and described physically. I see this as one of the most interesting results in this paper, and also novel, but if not explained in comparison to existing literature, it will fail to promote more research or will fail to inspire a paradigm shift."

We thank the reviewer for the comment.

Author reply: In the SI, even in the first revision, we had already included the physics behind the trend observed in our study. I copied the material from SI here.

Supplementary Figure 20. The nucleation temperature of water droplets, when the membrane is filled with water without addition of oil, measured through FTIR metrology is shown as a function of length scale.

Supplementary Figure 21. (a) The role of interfacial curvature on surface function and dependence of chemical potential difference on length scale is shown when pores are filled with water without addition of the oil. (b) In this scenario, the Gibbs energy barrier for water-ice phase change of nanodroplets is shown as a function of the diameter of nanodroplets.

“Ice nucleation temperature as a function of pore dimension is shown in **Supplementary Figure 20**. Note that this is completely in contrast to the case that the oil-water interface exists in which ice nucleation temperature drops at lower pore dimensions.

The observed nucleation temperature could be explained through the thermodynamics of ice nucleation. In this case, nanodroplets experience negative pressure inside the pores due to the curvature of the water meniscus. Surface factor, $f(m, x)$, and chemical potential difference, $\Delta\mu$, are calculated for this scenario as a function of pore dimension and are plotted in **Supplementary Figure 21a**. As shown, as the size decreases, the surface factor decreases as well, primarily due to the concave interface of pore walls. In addition, the negative Laplace pressure in the nanodroplet decreases, which in turn leads to an increase in the chemical potential difference. The decrease in $f(m, x)$ and the increase in $\Delta\mu$ both lead to drops in the Gibbs energy barrier for ice nucleation, **Supplementary Figure 21b.**”

The work by the Marcolli, 2017 is focused on pre-activation of pores for ice nucleation and growth. In our study, we filled the pores with water at room temperature and reduced temperature of the system in quasi-static approach until we observed freezing. Please look at the temperature of the experiments in **Supplementary Figure 19** which starts at 5 °C and is reduced in a step-wise manner. We **did not induce** any pre-activation condition and even if pre-active sites existed, it would be melted at room temperature. *As written in Marcolli 2017*, “Pre-activation is greatly enhanced when the temperature falls below the homogeneous ice nucleation threshold (Higuchi and Fukuta, 1966) and it is lost for $T > 273$ K with one exception reported by Edwards et al. (1970)”.

On the opposite trend, we think that the negative pressure of the liquid phase and its role on chemical potential difference (Eq. 2 in the manuscript as written below and Fig. S21a (above)) is the main cause of the opposite trend.

$$\Delta\mu_{Nd}(T, P) = \Delta\mu_b(T, P_{atm}) + (P_{Nd} - P_{atm})(v_w - v_i)$$

Action: We added few sentences in the manuscript and referenced Marcolli 2014.

“We should add that for fully water-filled pores, ice nucleation temperature increases with the pore dimension as discussed by Marcolli⁴⁹. The difference in the trend compared to our work could be due to extremely low pressure (negative pressure) of studied droplets here and its consequent effect on the chemical potential difference, see **Supplementary Note 8.**”